# Multi-ancestry and multi-trait genome-wide association meta-analyses inform clinical risk prediction for systemic lupus erythematosus

Chachrit Khunsriraksakul [1,2], Qinmengge Li[3], Havell Markus[1,2], Matthew T. Patrick[3], Renan Sauteraud[4], Daniel McGuire[4], Xingyan Wang[4], Chen Wang [1], Lida Wang [4], Siyuan Chen[4], Ganesh Shenoy[5], Bingshan Li [6], Xue Zhong[7], Nancy J. Olsen[8], Laura Carrel[9], Lam C. Tsoi[3,10], Bibo Jiang[4,10] & Dajiang J. Liu [1,2,4,10] ✉

Systemic lupus erythematosus is a heritable autoimmune disease that predominantly affects young women. To improve our understanding of genetic etiology, we conduct multi-ancestry and multi-trait meta-analysis of genome-wide association studies, encompassing 12 systemic lupus erythematosus cohorts from 3 different ancestries and 10 genetically correlated autoimmune diseases, and identify 16 novel loci. We also perform transcriptome-wide association studies, computational drug repurposing analysis, and cell type enrichment analysis. We discover putative drug classes, including a histone deacetylase inhibitor that could be repurposed to treat lupus. We also identify multiple cell types enriched with putative target genes, such as non-classical monocytes and B cells, which may be targeted for future therapeutics. Using this newly assembled result, we further construct polygenic risk score models and demonstrate that integrating polygenic risk score with clinical lab biomarkers improves the diagnostic accuracy of systemic lupus erythematosus using the Vanderbilt BioVU and Michigan Genomics Initiative biobanks.

Systemic lupus erythematosus (SLE) is a chronic multi-organ autoimmune disease that predominantly affects young women and individuals of African, Asian, and Hispanic ancestries[1,2]. The prevalence of SLE worldwide was estimated to range from 37 to 123 cases per 100,000 individuals depending on geographic locations, ancestry groups, study type, and study period[3]. No cure or targeted treatment exists for SLE. Current medications for treatments are broadly acting (e.g., corticosteroid) and are associated with many side effects[4]. The diagnosis of SLE is challenging due to the heterogeneity in clinical symptoms and lack of pathognomonic features or accurate lab tests[5]. The difficulty in the diagnosis of SLE presents great challenges for treatment, as advanced-stage SLE is associated with a worse prognosis and can lead to organ failure and even deaths[6]. Thus, accurate early detection and intervention are the key to mitigating SLE disease outcomes and improving quality of life[7].

[1]Program in Bioinformatics and Genomics, Pennsylvania State University College of Medicine, Hershey, PA 17033, USA. [2]Institute for Personalized Medicine, Pennsylvania State University College of Medicine, Hershey, PA 17033, USA. [3]Department of Dermatology, University of Michigan Medical School, Ann Arbor, MI 48109, USA. [4]Department of Public Health Sciences, Pennsylvania State University College of Medicine, Hershey, PA 17033, USA. [5]Department of Neurosurgery, Pennsylvania State University College of Medicine, Hershey, PA 17033, USA. [6]Department of Molecular Physiology & Biophysics, Vanderbilt University, Nashville, TN 37235, USA. [7]Department of Medicine, Division of Genetic Medicine, Vanderbilt University Medical Center, Nashville, TN 37232, USA. [8]Department of Medicine, Pennsylvania State University College of Medicine, Hershey, PA 17033, USA. [9]Department of Biochemistry and Molecular Biology, Pennsylvania State University College of Medicine, Hershey, PA 17033, USA. [10]These authors jointly supervised this work: Lam C. Tsoi, Bibo Jiang, Dajiang J. Liu. ✉e-mail: dajiang.liu@psu.edu

Genome-wide association studies (GWAS), functional genomic studies, and integrative analysis have provided unprecedented understanding of the genetic architecture underlying complex diseases[8]. Such analyses have enabled the identification of candidate causal genes or tissues/cell types and the development of robust bio-markers and targeted drugs[9]. GWAS have also been successful in SLE[10]. To date, GWAS have identified >130 loci that reach genome-wide significance (*P* value $< 5 \times 10^{-8}$)[11-14]. Yet, these identified loci explain only a small fraction of overall heritability and the genes or tissues/cell types affected by them remain unclear[11,15]. Previous SLE GWAS results were primarily from individuals of European ancestry[16-19]. Fortunately, several recent GWAS efforts have greatly increased the sample size of non-European ancestry including East Asians[11,20-23].

To take advantage of existing datasets and maximize sample sizes and power, we assembled multi-ancestry SLE GWAS datasets from European, Eastern Asian, and Admixed American ancestries. To exploit the shared genetic basis between different autoimmune diseases, we also aggregated GWAS datasets from 10 genetically correlated auto-immune diseases. We conducted multi-ancestry and multi-trait meta-analysis. To gain mechanistic insights, we conducted transcriptome-wide association studies (TWAS), linking regulatory variants to their target genes. We leveraged computational drug repurposing (CDR) and cell type enrichment analyses to identify novel drugs with the potential to treat SLE as well as to identify previously described immune cell types that are dysregulated in SLE. Lastly, we constructed polygenic risk scores (PRS) and validated the models in two independent biobanks: Michigan Genomic Initiative (MGI)[24] and Vanderbilt University Biobank (BioVU) and investigated the utility of PRS to improve the diagnosis of SLE when used in conjunction with clinical lab tests, e.g., anti-nuclear antibody (ANA) and anti-double strand DNA (anti-dsDNA).

## Results

### SLE GWAS meta-analysis

We provide an overview of study design and schematic workflow in Supplementary Fig. 1. Our compiled SLE GWAS dataset contains 12 cohorts from three ancestries [East Asian (EAS), $N_{EAS} = 194,435$ (5877 cases and 188,558 controls); European (EUR), $N_{EUR} = 520,311$ (14,355 cases and 505,956 controls); Admixed American (AMR), $N_{AMR} = 3720$ (1393 cases and 2327 controls)], with a total number of N = 21,625 cases and 696,841 controls. We carefully conducted quality controls to ensure the validity of the analyses, including checking sample over-laps, and manually examining Manhattan and quantile-quantile (QQ) plots for each study. With quality-controlled SLE datasets, we first performed a fixed effect meta-analysis within each ancestry using inverse-variance weighted meta-analysis method.

Next, as SLE has overlapping symptoms and shared genetic basis with other autoimmune diseases[25-27], we conducted multi-trait association analysis to improve power (Supplementary Data 1 and Supplementary Fig. 2). We first calculated the genetic correlations between SLE and 13 other autoimmune disorders in samples of European ancestry using linkage disequilibrium score regression (LDSC)[28,29] (Fig. 1). We identified ten autoimmune disorders (autoimmune thyroid disorder, Crohn's disease, celiac disease, multiple sclerosis, primary biliary cirrhosis, rheumatoid arthritis, Sjogren's syndrome, systemic sclerosis, type 1 diabetes, and ulcerative colitis) as significantly genetically-correlated traits with SLE (false discovery rate <0.05) (Fig. 1). We then performed multi-trait analysis across autoimmune disorders of significant genetic correlation with SLE using MTAG[30] in each ancestry separately and combined the results across ancestries using fixed effect meta-analysis.

We show Manhattan plot and Quantile-Quantile (QQ) plot for multi-ancestry and multi-trait meta-analysis (MAMT) for SLE in Fig. 2 and Supplementary Fig. 3, respectively. The meta-analysis results show well calibrated QQ plot with genomic control value 1.03. If we define a locus to be a 1 million basepair window surrounding the sentinel variants, the meta-analysis identified 106 loci that reach genome-wide significance (*P* value $< 5 \times 10^{-8}$). Among the identified loci, 27 are novel that reach genome-wide significance for the first time. Alternatively, we may also define a locus based on the linkage disequilibrium (LD)-based pruning. We created a LD reference panel using a subset of 1000 Genomes Project phase 3 data, with the same ancestry fraction as

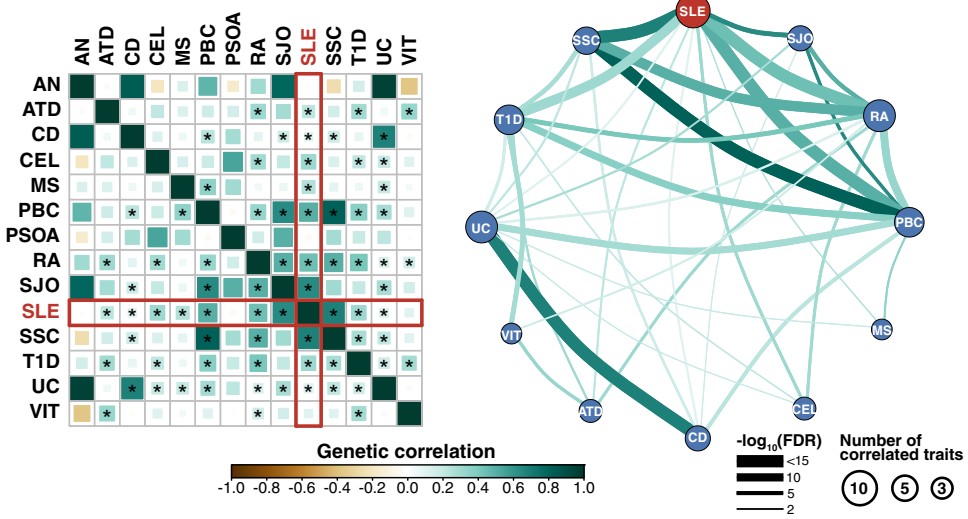

**Fig. 1 | Genetic correlations across 14 autoimmune diseases in European ancestry.** Left panel indicates genetic correlations across 14 autoimmune diseases. Genetic correlation is estimated using cross-trait LDSC. Color intensity and size of square are proportional to strength of genetic correlation (brown = negative correlation, green = positive correlation). Asterisks indicate genetic correlations that are statistically significant at false discovery rate level of 0.05. Red boxes highlight the correlation between SLE and other traits. Right panel illustrates the network of significantly genetically-correlated traits. The colors of the lines represent the magnitude of genetic correlation estimates using LDSC. The widths of the lines represent the statistical significance. Disease name abbreviations: AN = ankylosing spondylitis, ATD = autoimmune thyroid disease, CD = Crohn's disease, CEL = celiac disease, MS = multiple sclerosis, PBC = primary biliary cirrhosis, PSOA = psoriatic arthritis, RA = rheumatoid arthritis, SJO = Sjogren's syndrome, SLE = systemic lupus erythematosus, SSC = systemic sclerosis, T1D = type 1 diabetes, UC = ulcerative colitis, VIT = vitiligo.

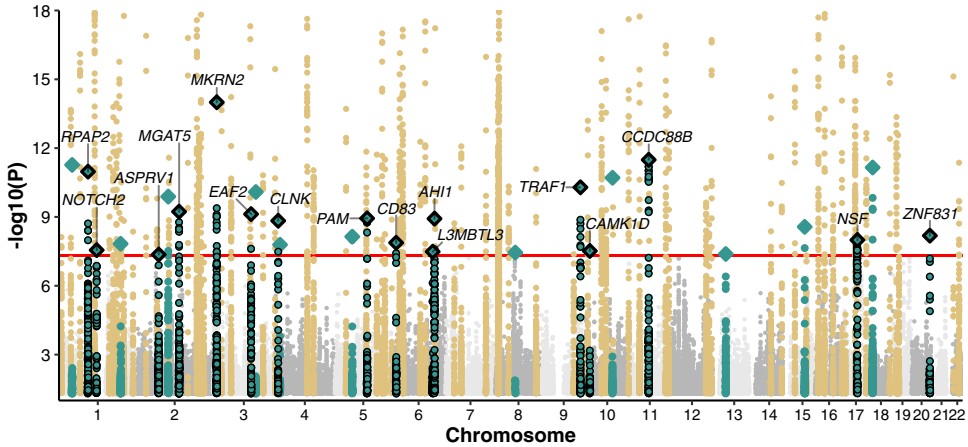

**Fig. 2 | Manhattan plot of the multi-ancestry and multi-trait SLE GWAS meta-analysis.** We consider loci reported in the GWAS catalog (as of 12/11/2021) or in the studies used in our meta-analysis as known (brown). We define novel loci (green) as the ones that are 1 million base pairs away from previously reported GWAS loci. Diamonds in the plot represent sentinel variants at novel loci. Diamond with black outline represents novel loci that are replicable according to RATES. In total, we identify 79 known loci and 27 novel loci, of which 74 known loci and 16 novel loci are deemed replicable (PPR > 0.90) by RATES. We also annotated the target genes for top variants at each novel locus using Open Targets Genetics database. Here, y axis is truncated at P value = $1 \times 10^{-18}$ to visualize the novel loci. The red line represents the genome-wide significance level ($5 \times 10^{-8}$). Two-sided P value associated with each variant is calculated according to the Chi-squared test statistic with 1 degree of freedom.

datasets in the meta-analysis. We performed LD-based pruning using PLINK with the created reference panel and the squared correlation between alleles ($r^2$) cutoff of 0.2. It should be noted that a total of 249 loci, of which 92 were considered novel, were identified when LD pruning was performed (**Methods** and Supplementary Data 2). For all subsequent presentations, we stick with distance-based loci.

We further assessed the replicability of each locus using MAMBA[31] and its extension RATES (**Method**), a statistically rigorous model-based method to assess replicability leveraging the strength and consistency of association signals across studies. The term replicability was introduced by McGuire et al. that refers to variants with genuine non-zero effects[31]. The posterior probability of replicability (PPR) quantifies how likely the signal is genuine and can be replicated in a sufficiently powered replication study, e.g., a large enough study from a matched population[31]. RATES results confirmed that 74 known loci and 16 novel loci were replicable with PPR > 0.90 (Supplementary Data 3). Across 16 novel and replicable loci, we found that multi-ancestry SLE-only GWAS already yielded genome-wide significant P values in 3 loci and borderline significant P values for the remaining 13 loci (highest P value = $8.19 \times 10^{-5}$). Importantly, rheumatoid arthritis's GWAS contributes the most to the identification of novel loci in SLE, having the smallest P values for 8 out of 16 loci in comparison to other autoimmune diseases (Supplementary Table 1). This is likely due to the largest sample size of autoimmune GWAS data and overlapping clinical features between SLE and RA[32]. Interestingly, all 16 novel loci do not yield significant P values using Cochran's Q tests for heterogeneities under the Bonferroni threshold of 0.05/106 (Supplementary Fig. 4 and Supplementary Data 4). As fixed effect meta-analysis favors loci with homogeneous effects, we further explored the extent of heterogeneity using additional sub-threshold variants ($P < 1.00 \times 10^{-6}$) and found that 97% of 144 loci do not show evidence of heterogeneity in effect sizes across different SLE studies with P values for Cochran's Q test ≥0.05/144. We noted that the variances of the effect size estimates from MAMT analysis are 59% smaller than multi-ancestry fixed effect meta-analysis (MA) for SLE, indicating that jointly analyzing multiple traits increases the effective sample sizes by 2.87-fold. 94 of previously known loci did not reach genome-wide significance in our study, yet most of them (91 out of 94 loci) still have association P values <0.05 (Supplementary Data 5). The reduced level of statistical significance for these loci may be due to spurious association, genetic effect

heterogeneity between studies, or because multi-trait analysis introduces noise for loci that are uniquely associated with one or few traits and reduces power.

## Linking GWAS hits to target genes

95% of identified sentinel variants are non-coding. 79% of variants in the identified loci (as defined by sentinel variants and variants having $r^2 > 0.8$ to sentinel variants) are non-coding. To interpret their functional consequence, we identified potential target genes of each top variant at novel locus using Open Targets Genetics database (Accessed 21 April 2022)[33]. There is clear biological relevance for the identified target genes (Fig. 2). For example, *MKRN2* was previously shown to regulate NF-$\kappa$B-mediated inflammatory response[34]. Another target gene *CCDC88B* is an important regulator of maturation and effector functions of T cell[35]. Glycosylation changes in T cells by *MGAT5* have been shown to impact T cell functions and are implicated in many autoimmune diseases[36,37]. Moreover, we identified *CD83* as another candidate target gene for SLE and studies have found soluble CD83 to be a promising therapeutic to interfere with autoimmunity in SLE[38]. We provide a description of the sentinel variant and targeted gene at each novel locus in Table 1.

## Transcriptome-wide association study implicates additional novel genes

We conducted transcriptome-wide association studies (TWAS). Specifically, we first derived gene expression prediction models using two reference datasets from disease-relevant tissues, i.e., the Genetic European Variation in Disease[39] (GEUVADIS; lymphoblastoid cell line (LCL); $n = 358$) and Depression Gene Network[40] (DGN; whole blood; $n = 873$) datasets. We used a new method PUMICE that integrates 3D genome and epigenetic information to improve prediction accuracy[41]. To assess the accuracy of prediction models, we calculate Spearman's correlation coefficients between measured and predicted gene expression using nested cross-validation as described in Khunsriraksakul et al.[41] and assess whether Spearman's correlation is significantly different from zero[41]. Using PUMICE, we obtained 7028 and 9260 significant genes with Spearman's correlation coefficients >0.1 and the corresponding P values <0.05 from GEUVADIS and DGN, respectively. Next, we conducted TWAS analysis using TESLA[42], which integrates our SLE GWAS summary statistics from multiple

**Table 1 | List of sentinel variants at sixteen replicable novel loci from the multi-ancestry and multi-trait meta-analysis**

| rsID | Chr:Pos (hg19) | Effect allele | Other allele | Beta | SE | P value | Mapped gene | PMID |
|---|---|---|---|---|---|---|---|---|
| rs299629 | 3:12576846 | A | G | −0.061 | 0.0078 | $1.0 \times 10^{-14}$ | MKRN2 | 28378844 |
| rs516124 | 11:64128423 | G | T | 0.069 | 0.0099 | $3.2 \times 10^{-12}$ | CCDC88B | 25403443 |
| rs6662618 | 1:92935411 | T | G | 0.083 | 0.0122 | $1.0 \times 10^{-11}$ | RPAP2 | 7962544 |
| rs3761847 | 9:123690239 | G | A | 0.051 | 0.0077 | $5.1 \times 10^{-11}$ | TRAF1 | 19433411 |
| rs13014122 | 2:135050622 | G | A | 0.051 | 0.0083 | $5.9 \times 10^{-10}$ | MGAT5 | 30538706 |
| rs12490565 | 3:121553719 | G | A | −0.054 | 0.0088 | $7.5 \times 10^{-10}$ | EAF2 | 19333917 |
| rs2288786 | 5:102600754 | G | A | 0.051 | 0.0083 | $1.1 \times 10^{-9}$ | PAM | 16107699 |
| rs9494331 | 6:136006301 | G | A | −0.064 | 0.0106 | $1.2 \times 10^{-9}$ | AHI1 | 16541099 |
| rs4697651 | 4:10721433 | C | T | 0.053 | 0.0088 | $1.5 \times 10^{-9}$ | CLNK | 10562326 |
| rs1535271 | 20:57734753 | G | A | 0.080 | 0.0120 | $6.5 \times 10^{-9}$ | ZNF831 | 34552111 |
| rs199533 | 17:44828931 | G | A | 0.066 | 0.0115 | $1.0 \times 10^{-8}$ | NSF | 25873919 |
| rs12529514 | 6:14096658 | T | C | −0.092 | 0.0162 | $1.3 \times 10^{-8}$ | CD83 | 23886695 |
| rs2453044 | 1:120508524 | A | G | −0.064 | 0.0115 | $2.8 \times 10^{-8}$ | NOTCH2 | 20531454 |
| rs6602588 | 10:12487996 | G | A | 0.044 | 0.0080 | $3.0 \times 10^{-8}$ | CAMK1D | 19815495 |
| rs6939565 | 6:130194204 | C | T | −0.042 | 0.0076 | $3.3 \times 10^{-8}$ | L3MBTL3 | 15889154 |
| rs12992553 | 2:70360262 | A | G | −0.046 | 0.0084 | $4.4 \times 10^{-8}$ | ASPRV1 | 29212956, 34766153 |

We use the reference allele as the effect allele, and report the effect size estimates, standard errors, P values, and mapped target gene based on Open Target database. We also provide key references that describe their biological functions related to SLE. Two-sided P value associated with each variant is calculated according to the Chi-squared test statistic with 1 degree of freedom.

participating studies of different ancestries with gene expression prediction models based on samples of European ancestry in DGN or GEUVADIS. Specifically, TESLA uses meta-regression to jointly model the genetic effects across ancestries so that we can borrow strength from shared effects between ancestries to optimize TWAS power (**Methods**). In total, we identified 99 and 119 significant gene-level associations using GEUVADIS and DGN trained models. We define loci iteratively: we first rank significant genes by their P values. For the most significant gene in the list, we define a locus as a 1 million basepair window surrounding the gene. We then remove all genes in the list that overlap the locus and repeat the process to define the next locus, until we exhaust all significant genes from the list. We define novel loci as the ones >1 million base pairs away from known GWAS sentinel variants. Using this criterion, we found 6 and 17 novel and independent loci from GEUVADIS and DGN analyses, respectively (Fig. 3a, b).

TWAS associations at 106 GWAS loci are reported in Supplementary Data 6 and 7. We further compared the TWAS results with the list of target genes linked to enhancer variants (enVars) via Open Target Genetics[43]. In total, we were able to link these enVars to 26 unique target genes and subsequently used these genes as a reference to benchmark the accuracy of our TWAS results. We found 22 enVar target genes overlapping TWAS genes with nominal significance levels, and 9 enVar target genes overlapping TWAS hits with significant P values under the Bonferroni threshold for testing multiple genes across the genome.

We identified many novel associations that are supported by biological links to SLE pathogenesis. For example, CD52 mRNA expression is elevated in B cells of SLE patients and may function in B cell homeostasis by inhibiting responses to B cell receptor (BCR) signaling[44]. Circulating IL8 is also elevated in SLE patients[45]. IL8 can activate neutrophils, leading to the release of neutrophil extracellular traps (NETs), which are implicated in many immune-mediated conditions[46]. UBASH3A is a suppressor of T cell receptor (TCR) signaling and is implicated in multiple autoimmune diseases[47–50]. Moreover, UBASH3A mRNA expression levels in peripheral blood mononuclear cells are decreased in SLE patients[51].

### Cell type enrichment analysis in DGN-trained TWAS associations

To identify immune cell types relevant for SLE, we conducted cell type enrichment (CTE) analysis to evaluate if cell type specific genes are enriched with TWAS associations. We provide more detailed description of the method and datasets in **Methods**.

In the Database of Immune Cell Expression (DICE) dataset, we observed significant enrichment of TWAS hits in various immune cell subsets, including non-classical monocytes, B cells, NK cells, CD8 + T cells, and $T_H1$ cells, whose dysregulation have been implicated in SLE (Fig. 3c). This observation is consistent with previous studies. For example, non-classical monocytes are increased in SLE patients compared to healthy controls. They tend to have inflammatory characteristics with properties of antigen presentation that activate T and B cells[52,53]. Naïve B cells was observed to be decreased in SLE patients when compared to patients with other autoimmune diseases and healthy controls[54]. In SLE patients, the chromatin accessibility of naïve B cells shows enrichment for transcription factors that lead to B cell activation (NFKB, AP-1, BATF, IRF4, and PRDM1)[55]. The role of NK cells remains unclear in SLE, as different subsets have shown to have either pro- or anti- inflammatory activities[56,57]. Similarly, although the role of naïve CD8+ T cells is unknown in SLE, their effect on the differentiation into activated CD8+ T cells has been studied. Activated CD8+ T cells in the peripheral blood of SLE patients are observed to have decreased effector and cytolytic function which likely contribute to autoimmunity[58,59]. Lastly, the ratio of $T_H1/T_H2$ are observed to be increased in peripheral mononuclear cells of SLE patients[60] along with dysregulation of $T_H1$ cytokines, such as IFN-γ[61].

We also carried out a second cell-type enrichment analysis, focusing on B cell subtypes collected from SLE patients[62]. This is because one of the hallmarks of SLE is the production of autoantibodies, which are due to autoreactive B cells. We observed significant enrichment of TWAS signals in cell type specific genes of double negative B cells, transitional 3 B cells, activated naïve B cells, and isotype switched memory B cells, whose dysregulation has been implicated in SLE etiology (Fig. 3c). For example, Scharer et al. observed all B cell subtypes were distinct between healthy controls and SLE[62]. Double negative B cells are likely to differentiate into antibody secreting cells. Their transcriptomes are more closely related to activated naïve B cells and isotype switched memory B cells in SLE samples than in healthy control cells[62]. Overall, our analysis highlights dysregulation of specific immune cells subsets that are previously supported to be aberrant in SLE, which could serve as potential targets for drug development and immune therapy.

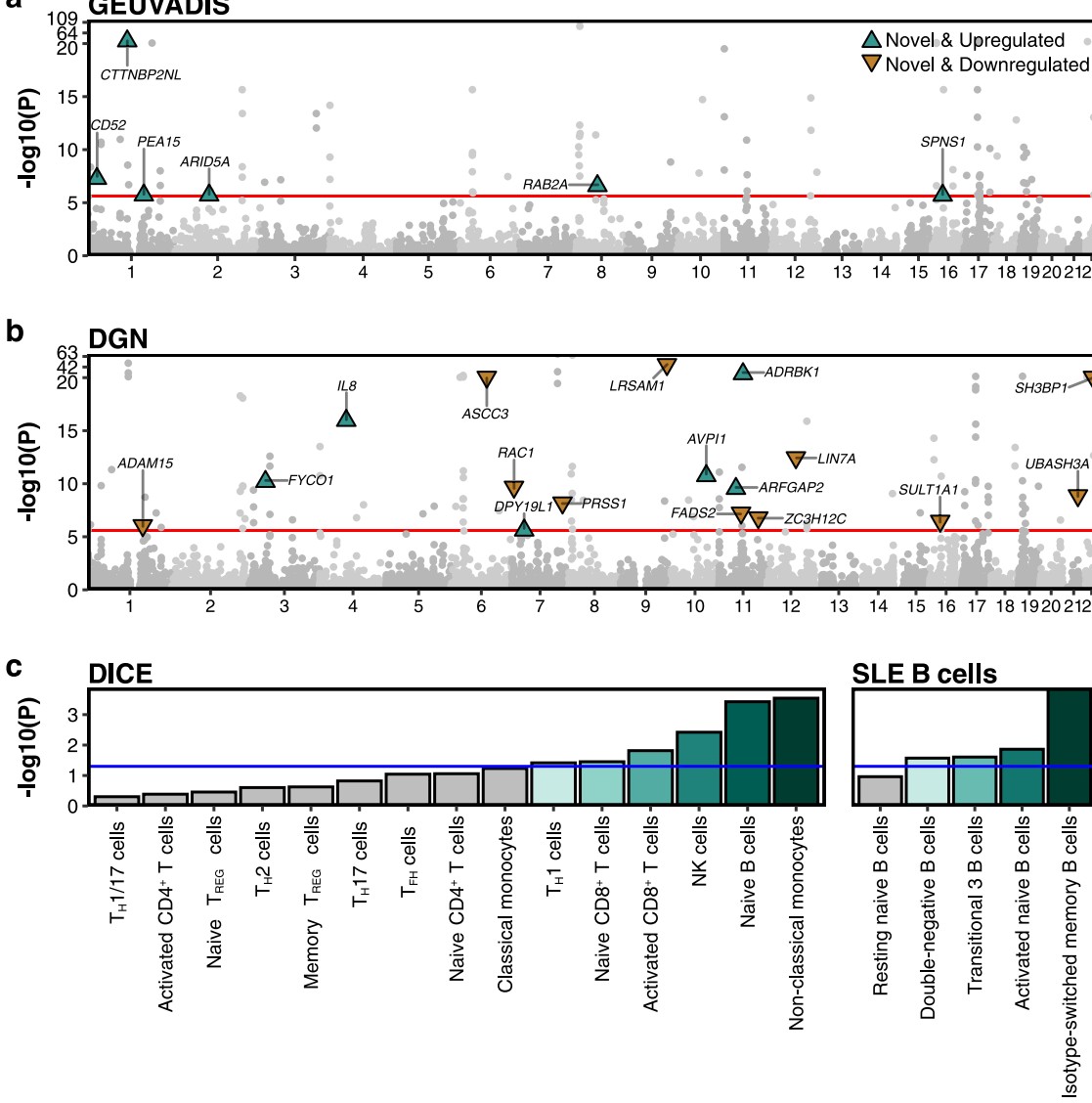

**Fig. 3 | Manhattan plot of the transcriptome-wide association studies and subsequent cell type enrichment analyses.** Gene expression prediction models are trained in either lymphoblastoid cell line (GEUVADIS dataset, panel **a** or whole blood (DGN dataset, panel **b**. The red horizontal line represents the significance threshold at $2.5 \times 10^{-6}$ (Bonferroni threshold for testing 20,000 genes). We label the most significant gene at each novel locus outside identified GWAS loci, which we defined as ±1 Mb region surrounding the sentinel variants. Two-sided $P$ value associated with each gene is calculated based on the TWAS Z score for gene-based association test. Panel **c** illustrates the results for cell type enrichment analyses of TWAS signals from whole blood (DGN) for 15 immune cell types from DICE dataset (left panel) and 5 B cell subsets from SLE patients (right panel). We assess the $P$ value of enrichment using a Gaussian copula-based multiple regression model as described in Chen et al.[42]. The blue horizontal line represents the significance threshold at 0.05. Bar plots highlighted in green colors represent the significant cell type enrichment (two-sided $P$ value <0.05).

## Computational drug repurposing analysis identified novel therapeutics for SLE

We perform computational drug repurposing (CDR) via Connectivity Map (CMap)[63,64] using significant gene targets identified in TWAS associations. Importantly, many clinically informative drug classes were identified, including glucocorticoid receptor agonist, histone deacetylase (HDAC) inhibitor, mTOR inhibitor, and topoisomerase inhibitor (Fig. 4). Some of the identified drug classes, including glucocorticoid receptor agonists, are already being commonly used in the clinic to treat SLE and supports the validity of these results. An HDAC6 inhibitor was previously shown to greatly reduce lupus nephritis in mice[65]. An mTOR inhibitor was also shown to attenuate SLE by regulating inflammation induced CD11b+ Gr1+ myeloid cells[66]. Importantly, sirolimus, an mTOR inhibitor, has already been shown to improve SLE disease activity in Phase 1/2 clinical trial[67]. Irinotecan, a topoisomerase

inhibitor, also reverses lupus nephritis and results in prolonged survival in SLE mice[68].

## Derivation of PRS models

We derived multiple candidate PRS models using a number of methods, including pruning and thresholding (P+T), SBayesR[69], SBLUP[70], SDPR[71], LDpred-inf[72], LDpred-funct[73], PUMAS[74], PRS-CS-auto[75], and LASSOSUM[76]. We only used summary statistics-based PRS methods, which do not require training or replicating cohorts to select tuning hyperparameters. We analyzed three sets of GWAS summary statistics, including 1) SLE GWAS of European ancestry ($N = 6748$ cases and 11,516 controls)[17] (Ref), which we also include as part of MA and MAMT analyses, 2) MA for SLE, and 3) MAMT for SLE. To validate the accuracy of PRS models, we use two independent biobanks that are linked to electronic medical records

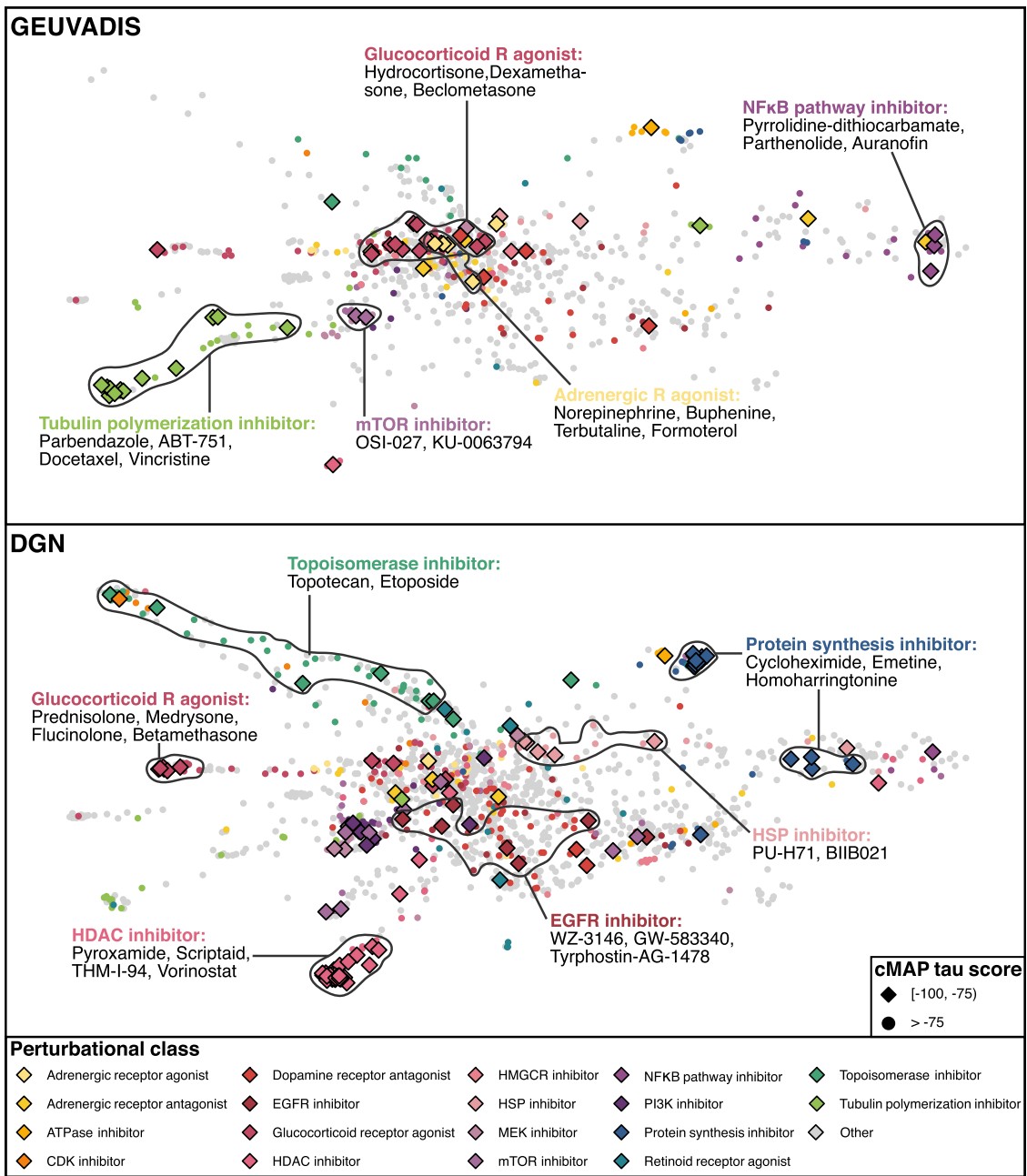

**Fig. 4 | Computational drug repurposing analysis via Connectivity Map (CMap).** We used significant SLE TWAS associations (P value <2.5 × 10⁻⁶; two-sided P value associated with each gene is calculated based on the TWAS Z score for gene-based association test) in CMap to identify drugs that could potentially reverse the disease signature. Data are visualized with L1000FWD. Each point represents cell-specific drug-induced signature, and signatures were clustered via k-nearest neighbors. To quantify the correlation between a query signature and reference profile, CMap calculates a τ score. A negative τ score indicates that trait-associated gene expression profile will be normalized by the identified molecule, which we can potentially repurpose to treat the disease. A CMap τ score of −75 indicates that the drug more consistently reverses the expression level of the TWAS significant genes than 75% of all reference gene sets. Diamond shape represents signatures with CMap τ scores less than −75. Signatures are colored by perturbational drug classes. Contour lines and labels represent frequently identified drug class and associated drugs.

(EMRs), including MGI (N = 34,702) and BioVU (N = 49,707), which are not part of the GWAS discovery cohorts. Since biobank data may define SLE cases differently from the discovery cohorts, we performed GWAS meta-analysis of BioVU and MGI datasets. We confirmed that the effect size (beta) between the biobank datasets and GWAS discovery cohorts have good concordance with each other (Pearson correlation coefficient of 0.7 and P value = 4.4 × 10⁻¹⁶) (Supplementary Fig. 5).

We calculated the proportion of variance explained on the liability scale (R²) and area under the receiver operating characteristic curve (AUC) for each PRS model in the MGI and BioVU datasets and selected the best PRS model for subsequent downstream analyses (Supplementary Fig. 6). As MGI and BioVU are predominantly of European ancestry, we focus on samples of European ancestry in our analyses. We provided other performance metrics, including true negative, true positive, false negative, false positive, sensitivity, specificity, odd ratio (OR) per standard deviation, OR of top 20% vs. bottom 20% and its 95% confidence interval, and Nagelkerke's R² and its 95% confidence interval (NKR2 [95% CI]) in Supplementary Data 8.

We found that LASSOSUM on average yielded more accurate models compared to other PRS methods (Supplementary Data 9). LASSOSUM risk scores generated an AUCs of 0.75 and 0.74 in MGI and BioVU, respectively. Moreover, PRS based on GWAS summary statistics derived from MAMT almost always performed better than that of MA or Ref regardless of the PRS methods used. We also provide the number of variants used as predictors in each PRS model in Supplementary Table 2.

## Sensitivity analyses of PRS models

Since our external testing datasets are EMR-based biobanks, which are susceptible to incorrect data entry or false positive diagnosis, we conducted sensitivity analyses to evaluate how the PRS models performed using SLE cases defined by three different algorithms. A list of ICD codes used in this study can be found on Supplementary Table 3. Specifically, we utilized algorithms previously evaluated by Barnado et al.[77]. We considered three different algorithms listed on Supplementary Table 4, including

- Def1 (least stringent) that only requires at least one count of SLE ICD codes;
- Def6 (intermediate) that requires at least two separate counts of SLE ICD codes, and excludes patients with systemic sclerosis or dermatomyositis ICD codes, and
- Def12 (most stringent) that requires at least four separate counts of SLE ICD codes and a recorded ANA positive test result (≥1:160). Moreover, patients with systemic sclerosis or dermatomyositis ICD codes are excluded.

Characteristics of patients identified from different algorithms and biobanks, including sample size, age at first diagnosis, ANA and anti-dsDNA status, and the number of patients with undifferentiated connective tissue disease, can be found on Supplementary Table 5. Importantly, the use of different definitions yielded comparable AUCs of PRS. When defining SLE cases using more stringent criteria (i.e., using Def 12 instead of Def 1), we improved the AUC of PRS from 0.75 to 0.78 in MGI and 0.74 to 0.79 in BioVU (Supplementary Fig. 7).

## Integrating PRS with conventional lab tests led to further improvement in diagnostic accuracy

Currently, the diagnosis of SLE follows the 2019 EULAR/ACR classification criteria[78]. To be considered as SLE, patients must have positive ANA tests (titer ≥1:80) and must tally at least 10 total points from clinical domains (e.g., malar rash) and immunologic domains (e.g., test positive for anti-dsDNA). The biomarkers ANA and anti-dsDNA alone have low accuracy and will only be prescribed when symptoms are already present.

We hypothesized that PRS for SLE can stratify SLE patients, improve the accuracy of conventional lab tests, and facilitate early diagnosis. We first assessed the utility of PRS in stratifying individuals for the risk of developing SLE using two independent biobanks, i.e., BioVU and MGI[24]. We observed that there were 3.81 times as many cases in the top quintile than in the bottom quintile of the PRS distribution (OR [95% CI]; MGI: 3.46 [2.36, 5.08], BioVU: 4.74 [3.59, 6.25]) (Fig. 5a).

We, then, evaluated the model performance of PRS when used in conjunction with ANA/anti-dsDNA lab results. We found consistent improvement in prediction accuracy when we added PRS into the prediction model. Specifically, in BioVU, the AUC of using PRS + ANA + anti-dsDNA is 0.75, which improves the AUC of using only ANA + anti-dsDNA (0.73) with P value 0.005 (Fig. 5b). Similarly, in MGI, the AUC of using PRS + ANA + anti-dsDNA is 0.75, which improves the AUC of using ANA + anti-dsDNA alone (0.74) with P value of 0.065. When jointly analyzing the MGI and BioVU cohorts, the AUC improves from 0.72 with ANA + anti-dsDNA alone to 0.74 with PRS + ANA + anti-dsDNA (P value = 0.002).

Lastly, we investigated the capability of PRS to stratify patients with certain classical SLE lab results. Among individuals that are ANA positive or anti-dsDNA-negative, the top PRS quintile has approximately 2.36 and 2.34 times more SLE cases than in the bottom PRS quintile, respectively (Fig. 5c). When focusing on patients who had positive ANA and negative anti-dsDNA tests in the MGI and BioVU biobanks (the most common lab test results observed clinically), the top PRS quintile has approximately 2.31 times more SLE cases than in the bottom PRS quintile. Importantly, the risk gradient curve showed a good stratification of positive ANA and negative anti-dsDNA patients, especially, in BioVU database (prevalence of SLE in bottom quintile is 9.91%, middle quintile is 15.09%, and top quintile is 28.88%). The top quintile has 2.91 and 1.91 times more SLE cases than those in the bottom and middle quintiles, respectively. This result shows that PRS can help increase the specificity of ANA test and sensitivity of anti-dsDNA test.

## Discussion

In this work, we carried out multi-ancestry and multi-trait GWAS meta-analysis of SLE by aggregating the GWAS results from 12 SLE cohorts, encompassing over 700,000 samples. In total, we identified 106 significant loci, of which 16 loci were deemed as novel and replicable. We further discovered 22 additional novel loci through TWAS and performed computational drug repurposing and cell type enrichment analyses. Our results pointed out a few relevant drug classes, such as glucocorticoid receptor agonist, HDAC inhibitor, and mTOR inhibitor, as well as enriched cell types, such as non-classical monocytes and B cells. The identified drugs are capable of reversing SLE-associated gene expression signatures and are therefore putative candidates for SLE treatment. The roles of these enriched cell types in SLE are also supported by previous experimental evidence.

Based on the meta-analysis results, we applied nine methods to derive PRS models. We assessed the accuracy of the PRS models in two independent EMR-based biobanks, including MGI and BioVU, and evaluated their utility in identifying SLE patients when used in conjunction with conventional lab tests. We found that PRS of SLE was highly effective for identifying patients with an elevated risk of SLE, as there were 3.81 times as many cases in the top PRS quintile than in the bottom PRS quintile. Moreover, we showed that PRS can help increase the specificity of ANA test and the sensitivity of anti-dsDNA test. The results in MGI and BioVU datasets slightly differ, which may be due to factors such as geographic catchment and sampling strategy[24]. Nonetheless, the results are, in general, concordant with each other and demonstrate the added utility of PRS in diagnosing SLE. Our results demonstrate the potential benefits of incorporating PRS into SLE classification criteria, which can lead to a more accurate early diagnosis for SLE.

Our work helps establish the added benefits of PRS for improved diagnosis when used together with conventional lab tests. Earlier studies have applied PRS in various health outcomes including type 2 diabetes, cardiovascular disease, and others[79-82]. However, the added benefits of PRS, when used with other diagnostic criteria and lab tests, remain under-explored, and completely unexplored to our knowledge for SLE. Our work showed that PRS when used in conjunction with conventional lab values such as ANA and anti-dsDNA status, could improve diagnostic accuracy. PRS would not replace conventional lab tests, but as germline DNA usually would not change over a lifetime, the use of PRS could facilitate early diagnosis and risk screening. PRS has the most clinical utility in individuals with extreme PRS values, who will benefit from careful monitoring for progression[83]. With the improvement in the understanding of the polygenic architecture of complex diseases, the advancement of statistical methodology, and the increase of genetic diversity in genotyped samples, we could expect further improvements in the accuracy of genetic prediction[83].

 

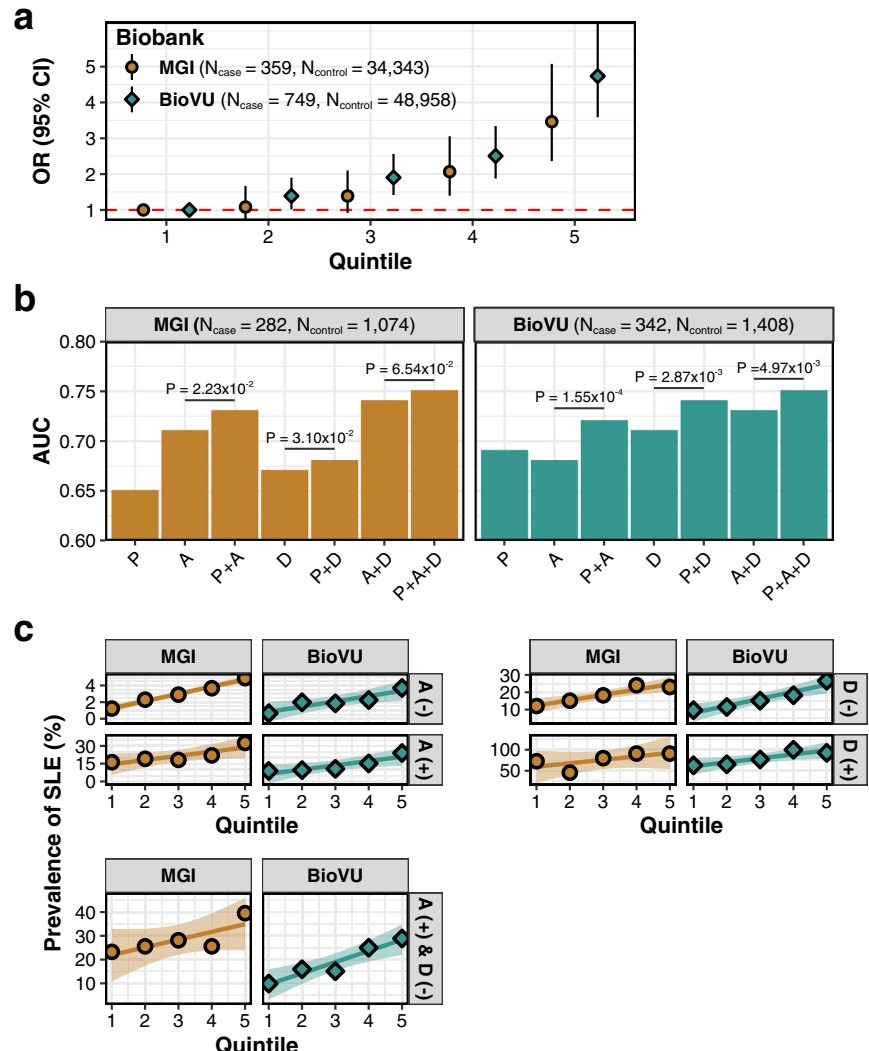

**Fig. 5 | Applications of PRS model derived from LASSOSUM using multi-ancestry and multi-trait GWAS data. a** The odd ratios (OR) and 95% confidence intervals (CIs; bars) of each PRS stratus against individuals in the lowest quintile. We stratified PRS into 5 groups with the bottom quintile used as the reference group. The red dashed line indicates OR = 1. **b** AUC of PRS (P), ANA (A), anti-dsDNA (D), and when these tests are used together to diagnose SLE. P values are calculated via one-sided Delong's test. It should be noted that this analysis is done in a subset of patients who have available information on both ANA and anti-dsDNA tests. **c** Risk gradient curves for patients with positive/negative ANA test results, positive/negative anti-dsDNA test results, and positive ANA and negative anti-dsDNA test results. The lines represent the relationship between PRS quintiles and fractions of SLE cases, and the shaded areas surrounding the lines represent 95% CIs. Here, SLE cases are defined according to Def1 (Supplementary Table 4).

Several future directions in data generation and method development could further enhance the predictive capability of PRS model in diagnosing SLE. First, since SLE is an autoimmune disease with a relatively low prevalence, increasing GWAS sample sizes will further improve marginal effect estimates and in turn, improve the predictive value of PRS. Second, even though SLE is more prevalent in non-European ancestry compared with Caucasians[84], 65% of the SLE GWAS samples in our study were from European ancestry. It is imperative to increase the diversity of SLE GWAS in the next phase of studies. Lastly, it remains an open question of how to best combine multi-ancestry genetics data to create a better and more transferable PRS model for different ancestries. It will be of great interest to investigate whether PRS similarly improves the diagnosis for SLE in individuals of non-European ancestry when used together with conventional lab tests. The utility of PRS in samples of European and East Asian populations will help motivate the development and deployment of PRS models in other populations.

In summary, our work represents one of the first attempts of applying PRS and clinical lab tests to enhance the early diagnosis of SLE. We can apply the framework to study other diseases to identify high-risk individuals from the population and facilitate the practice of targeted treatment and precision medicine.

## Methods

### Quality control of GWAS summary statistics

A list of all GWAS for 14 autoimmune diseases used in this study was listed in Supplementary Data 1. We retrieved GWAS summary statistics from GWAS catalog[85], FinnGen website (Release 5 data), Pan-UK Biobank website, and BioBank Japan PheWeb[22].

Prior to meta-analysis, we conducted careful quality control including examining the genomic control values, and manually inspecting the QQ plot and Manhattan plot from participating studies. We also harmonized GWAS data from participating studies and calibrated the effect allele to the reference allele. We also ensured that participating studies have little, or no sample overlaps and the effect

sizes are homogeneous between studies of similar ancestry, by calculating the $\lambda_{meta}$ statistic[86] (Supplementary Fig. 8). Specifically, for a SNP $i$ with estimated effect size $b_{i,1}$ and $b_{i,2}$ and variances $\sigma_{i,1}^2$ and $\sigma_{i,2}^2$ in a pair of cohorts 1 and 2, we calculated a statistic $T_i = \frac{(b_{i,1}-b_{i,2})^2}{\sigma_{i,1}^2 + \sigma_{i,2}^2}$, and denote the vector of statistics for all $M$ SNPs as $\mathbf{T}=(T_1,\ldots,T_M)$. $\lambda_{meta}$ is then calculated as $\frac{median(\mathbf{T})}{median(\chi_1^2)}$. If $\lambda_{meta}$ is much greater than 1, it suggests that there may be heterogeneity between estimated genetic effects from cohorts 1 and 2. If $\lambda_{meta}$ is much smaller than 1, it suggests that there may be overlapping samples. We have ensured that all pairs of studies have $\lambda_{meta}$ values close to 1 (Supplementary Fig. 8). For studies with known sample overlaps, we de-correlated the association statistics to ensure valid meta-analysis results[87].

For the situation where one study is a multi-ancestry meta-analysis and included all of the samples from another study (usually occur in multi-ancestry GWAS), we subtracted the GWAS summary statistics of the smaller study from the larger study according to the inverse-variance weighted meta-analysis formula[88], so that we can have non-overlapping studies for ancestry-specific analysis.

Previous studies observed that PRS model derived from HapMap3 SNPs have comparable performance to models that uses SNP from 1000 Genomes Project[89]. As such, we used SNPs in HapMap3 to construct PRS.

## Meta-analysis and annotation of the genome-wide significant variants

For MA, we used inverse-variance meta-analysis across 12 SLE cohorts as implemented in the METAL software[88]. For MAMT, we first analyzed each disease within each ancestry using the inverse-variance weighted fixed effect meta-analysis method in METAL[88]. Next, we utilized meta-analysis results from European ancestry to calculate genetic correlations among 14 autoimmune diseases using the LDSC software[28,29], as samples of European ancestry have the largest sample sizes. We considered traits to be genetically-correlated if genetic correlation P values (two-sided) are significant after controlling false discovery rate (FDR) at 0.05 level. We then performed multi-trait analysis combining SLE and its significantly correlated traits in each ancestry separately using the MTAG software[30]. Lastly, we combined multiple trait analysis results across ancestries using inverse-variance weighted meta-analysis[88]. HLA region was excluded from the multi-trait meta-analysis as was done for other autoimmune diseases due to their unusually large effect sizes that violate the model assumptions for MTAG. Cochran's Q test was performed via METASOFT[90] to estimate the heterogeneity of effect sizes across GWAS studies.

Genome-wide significance thresholds were defined as $5 \times 10^{-8}$. Independent genome-wide significant loci were defined as the ±1 Mb window surrounding the sentinel variant. We deem a locus novel if there were no previously reported significant variants in GWAS catalog or previous studies that fall within ±1 Mb from the sentinel variant. We then annotated potential target genes for sentinel variants according to the Open Target Genetics database, which used functional annotation data (e.g., eQTL, pQTL, and pc-HiC) to link regulatory variants to target genes. Alternatively, we also defined significant loci as the set of variants that are in LD ($r^2 > 0.2$) with the sentinel variants.

## Assessing the replicability of association signals

We assessed the replicability of association signals using a model-based approach. The method extends MAMBA[31] that uses a mixture model to assess whether the variants identified in a meta-analysis are genuine or spurious based on the strength and consistency of the association between studies. We define "replicable" variants as the ones with genuine association signals, so they can be replicated in a sufficiently powered replication study, e.g., one with large enough sample sizes from a matched population. Replicable signals with genuine non-zero effects tend to have stronger and more consistent signals across studies than spurious association signals. Here, to accommodate the potential heterogeneity of association signals between datasets of different ancestries, we use meta-regression with principal components of genome-wide allele frequencies as covariates and use the residuals as input for MAMBA analysis. RATES will assign a posterior probability of replicability to each signal which rigorously assesses its validity. We consider signals with PPR > 0.90 as strong evidence of replicability. We also visually inspected the Manhattan plot and forest plot for the SNPs deemed as replicable and confirmed the findings.

## Imputing gene expression prediction models and TWAS analysis

Gene expression prediction models were constructed via PUMICE[41]. Briefly, PUMICE utilizes epigenetic information to prioritize essential genetic variants that carry important functional roles. It also uses 3D genomic information to define windows that harbor cis-regulatory variants. Thus, PUMICE can more accurately predict gene expression levels using genotype data as input compared to alternative approaches. For epigenomic data, we utilize four broadly available epigenetic annotation tracks, including H3K27ac mark, H3K4me3 mark, DNase hypersensitive mark, and CTCF mark from ENCODE database[91,92]. Epigenomic data for whole blood and LCLs were retrieved from the ENCODE database with the following accession codes: ENCFF949VFY and ENCFF028SGJ, respectively. For 3D genomic data, we considered different choices for windows that harbor cis-regulatory variants, including the ones defined by conventional linear windows surrounding gene start and end sites (i.e., ±250 kb and ±1 Mb) as well as by 3D genomic informed regions (i.e., domain[93], loop[94], pcHiC[95], and TAD[94]). For whole blood (DGN), 3D genomic data from proxy tissue (lung for domain and spleen for loop, pcHiC, and TAD) is used. For LCLs (GEUVADIS), 3D genomic data from matched tissue is available. Model is deemed significant if the cross-validated average Spearman's correlation coefficient is >0.1 and the P value of the correlation coefficient is <0.05. We applied PUMICE to lymphoblastoid cell line (GEUVADIS) and whole blood (DGN) datasets as these tissues are most relevant to SLE's pathogenesis and have large sample sizes. For GEU-VADIS, we restricted our analysis to European samples (n = 358). For DGN, we only included samples with >90% European ancestry composition as determined from ADMIXTURE software using 1000 Genomes Project Phase 3 Data as reference panel (n = 873). With PUMICE, we obtained 7028 and 9260 gene expression models from GEUVADIS and DGN, respectively, for which the cross-validated average Spearman's correlation coefficients is >0.1 and the P value of the correlation coefficient is <0.05.

## Gene x trait association analysis with TESLA

The PUMICE prediction model is based on samples of European ancestry. We apply TESLA[42], a novel method to optimally integrate the prediction model with a multi-ancestry GWAS for TWAS. TESLA exploits shared phenotypic effects between ancestries and accommodates potential effect heterogeneities. TESLA is more powerful than alternative strategies that leverage fixed effect GWAS meta-analysis results to perform TWAS and the methods that only integrates only ancestrally matched GWAS and eQTL datasets.

## Cell type enrichment analysis

We retrieved cell type expression from two sources: 1) Database of Immune Cell Expression (DICE)[96] and 2) transcriptomic profiles of B cell subsets from SLE subjects[62]. Specifically, DICE profiled transcriptomic data of 15 immune cell types (2 of which are activated cell types) from 106 samples. For SLE-specific transcriptomic datasets, five B cell subtypes from nine SLE patients were profiled. We processed both datasets uniformly and closely followed the pipeline outlined in the previous study[97]. First, we normalized RNA-sequencing data in

terms of transcript per million (TPM). Next, we computed the average expression for each gene in each cell type. We removed genes not expressed across all cell types. We then rescale gene expression to a total of 1 million TPM for each cell type, to minimize the impact of library size. For each gene, we define the "gene expression specificity score" by dividing the expression of each gene in a given cell type by the total expression of the same gene across all cell types. We define the cell type-specific genes as the ones with gene expression specificity score in the top 5th percentile in each cell type.

To perform cell type enrichment analysis, we defined $Y$ as a vector of squared TWAS Z scores from $G$ significant genes and $\mathbf{X}$ as a matrix of $G$ rows and $C$ columns where rows correspond to genes and columns correspond to one of the $C$ cell types. The $(g, c)$ element of $\mathbf{X}$ is either 0 or 1, denoting whether gene $g$ is specific to cell type $c$ according to the definition of cell type specific gene above. As the genes may be correlated due to linkage disequilibrium, we calculate the correlation between squared TWAS Z score statistics via bootstrap. For the Z-score statistic from two genes 1 and 2, $Z_1$ and $Z_2$, the correlation between the TWAS Z-statistic can be calculated from LD panel as described in Khunsriraksakul et al.[41], which we denote as $\rho$. To calculate the correlation between squared Z statistics, we simulate bivariate normal distributed statistics $Z_1^{(b)}$ and $Z_2^{(b)}$, $b = 1,...,B$, where

$$Z_1^{(b)}, Z_2^{(b)} \sim N\left(\mathbf{0}, \begin{pmatrix} 1 & \rho \\ \rho & 1 \end{pmatrix}\right) \tag{1}$$

We then calculate the correlation between squared TWAS statistics as

$$\widetilde{\rho} = \frac{1}{B-1} \sum_{b=1}^{B} Z_1^{(b)} Z_2^{(b)} \tag{2}$$

We denote the correlation matrix for the squared Z statistics of $G$ genes as $\mathbf{P}$. We then performed weighted regression to assess whether cell type specific genes of a given cell type are enriched with TWAS hits. Specifically, we use

$$\mathbf{Y} = \mathbf{X}\boldsymbol{\gamma} + \boldsymbol{\epsilon} \tag{3}$$

We estimate the weighted regression coefficients as

$$\hat{\boldsymbol{\gamma}} = \left(\mathbf{X}'\mathbf{P}^{-1}\mathbf{X}\right)^{-1} \mathbf{X}\mathbf{P}^{-1}\mathbf{Y} \tag{4}$$

We assess the enrichment of each cell type by testing if the corresponding coefficient in $\hat{\boldsymbol{\gamma}}$ is significantly different from 0. This approach is similar in principle to MAGMA[98] and we adapt it for TWAS analyses.

## Computational drug repurposing

We extracted significant TWAS associations ($P$ value $<2.5 \times 10^{-6}$) and used them as a proxy for SLE disease signature. We then applied CMap algorithm to identify drugs capable of reversing disease signature. Specifically, we queried SLE TWAS association signals against the reference profiles in the CMap database (from L1000 assay), which recorded gene expression changes caused by perturbagens as the signature of the drug x gene pair. We only used reference data from touchstone dataset of CMap, which comprised reference signatures across nine cell lines treated with ~3000 well-annotated small molecule drugs. We consider a gene set to have stronger connectivity with the drug if the drug more consistently reverses the expression levels of all genes in the gene set. To quantify the correlation between a query signature and reference profile, CMap calculates a $\tau$ score. A negative $\tau$ score indicates that trait-associated gene expression profile will be normalized by the identified molecule, which we can potentially repurpose to treat the disease. $\tau$ score allows us to compare the

strength of connectivity between different gene sets. For example, CMap $\tau$ score of $-75$ indicates that the drug more consistently reverses the expression level of the TWAS significant genes than 75% of all reference gene sets. We consider a more negative $\tau$ score as stronger evidence that supports repurposing the drug for treating the disease. Finally, we adapted L1000FWD plot to visualize drug-induced transcriptomic signatures[99].

## Construction of PRS models

For our PRS model derivation, we applied nine PRS methods that do not require validation datasets for parameter tuning, including P+T, SBayesR[69], SBLUP[70], SDPR[71], LDpred-Inf[72], LDpred-funct[73], PUMAS[74], PRS-CS-auto[75], and LASSOSUM[76], to three GWAS summary statistics. We used 503 European samples from 1000 Genome Project Phase 3[100] as a reference panel for estimating linkage disequilibrium coefficients, per recommendations by previous work[89].

For the P+T method, we followed the recommendation from the Global Biobank Meta-Analysis Initiative[89]. Specifically, we ran PLINK[101] with the following flags: -clump-p1 1 -clump-p2 1 -clump-r2 0.1 -clump-kb 250. Next, we applied 13 different P value thresholds, including $5 \times 10^{-8}$, $5 \times 10^{-7}$, $1 \times 10^{-6}$, $5 \times 10^{-6}$, $5 \times 10^{-5}$, $5 \times 10^{-4}$, $5 \times 10^{-3}$, 0.01, 0.05, 0.10, 0.20, 0.50, 1. For other PRS methods, we used default settings.

## Study samples in MGI and BioVU

We first determined ancestry of each sample via ADMIXTURE[102] using 1000 Genome Project Phase 3 data as a reference panel. We only included samples with >90% European ancestry composition for subsequent analyses. Genetic data was imputed via Michigan Imputation Server using 1000 Genome phase 3 as the reference panel and only variants with imputation quality metric Rsq > 0.8 were kept for subsequent analyses.

We then performed quality control of the data following the recommendation by Marees et al.[103]. Specifically, with PLINK, we excluded 1) SNPs with the low genotyping rate (-geno 0.01), 2) individuals who have high rates of genotype missingness (-mind 0.01), 3) SNPs with low minor allele frequency (-maf 0.05), 4) SNPs that deviate from Hardy-Weinberg equilibrium (-hwe 1e-6), 5) individuals with high or low heterozygosity rates, 6) individuals that have a first or second-degree relative in the sample (-rel-cutoff 0.125), and 7) SNPs not within the HapMap3 SNP set.

Next, we extracted phenotypic data from the electronic medical records. Specifically, we used ICD codes to extract disease status (Supplementary Table 3). For ANA test result, we considered a titer of $\geq 1:80$ (e.g., 1:80, 1:160, 1:320, etc.) as positive, and the other values as negative (e.g., negative status, 1:40, 1:20, and etc.). For anti-dsDNA test result, we considered a level of $\geq 60$ IU/mL or positive status as positive, and the other values as negative (e.g., <60 IU/mL or negative status).

## Evaluation of prediction performance

We evaluated the prediction performance of constructed PRS in two independent biobank datasets: MGI and BioVU, which are not part of the training data. Specifically, we calculated Nagelkerke's $R^2$ on the liability scale[104] after adjusting for Sex, PC1-10, and HLA alleles[105] (HLA-DRB1*03:01, HLA-DRB1*08:01, and HLA-DQA1*01:02) that we imputed using HIBAG[106]. We assumed population prevalence of SLE to be 0.1%[107]. Furthermore, we also reported area under the receiver operating characteristic curve (AUC) for full model with the above covariates. We estimated the corresponding 95% confidence intervals from bootstrap with 1000 replicates. Comparison of AUCs between PRS models was calculated with two-sided Delong's test. Lastly, we divided the target samples into quintiles according to PRS rankings. We calculated odds ratios of developing SLE in each quintile against individuals with PRS in the bottom 20th percentile. Sensitivity and specificity were calculated at the optimal threshold using the Youden's J statistic.

## Software URLs

LDSC software can be found at https://github.com/bulik/ldsc. MTAG software can be found at https://github.com/JonJala/mtag. Open Target Genetics software can be found at https://genetics.opentargets.org. METASOFT software can be found at http://genetics.cs.ucla.edu/meta/. MAMBA software can be found at https://github.com/dan11mcguire/mamba. PUMICE software can be found at https://github.com/ckhunsr1/PUMICE. TESLA software can be found at https://github.com/funfunchen/rareGWAMA. CMap software can be found at https://clue.io/. L1000FWD web application can be found at https://maayanlab.cloud/L1000FWD. SBayesR software can be found at https://github.com/YinLiLin/hibayes. SBLUP software can be found at https://yanglab.westlake.edu.cn/software/gcta. SDPR software can be found at https://github.com/eldronzhou/SDPR. LDpred-Inf software can be found at https://privefl.github.io/bigsnpr/articles/LDpred2.html. LDpred-funct software can be found at https://github.com/carlaml/LDpred-funct. PUMAS software can be found at https://github.com/qlu-lab/PUMAS. PRS-CS-auto software can be found at https://github.com/getian107/PRScs. LASSOSUM software can be found at https://github.com/tshmak/lassosum. PLINK software can be found at https://www.cog-genomics.org/plink. Bedtools software can be found at https://bedtools.readthedocs.io/en/latest. ADMIXTURE software can be found at https://bioinformaticshome.com/tools/descriptions/ADMIXTURE.html. Michigan imputation server can be found at https://imputationserver.sph.umich.edu/index.html. R Project for statistical computing can be found at https://www.r-project.org.

## Reporting summary

Further information on research design is available in the Nature Portfolio Reporting Summary linked to this article.

## Data availability

The GWAS summary statistics of the multi-ancestry and multi-trait SLE meta-analysis result have been deposited on the Shiny App [https://liugroupstatgen.shinyapps.io/SLEv] for users to download and interactively explore research results. This meta-analysis result was derived via MTAG[30] and METAL[88] from the following datasets: SLE[11,16–23,108], ATD[22], CD[109,110], CEL[111,112], MS[112–114], PBC[115,116], RA[22,117], SJO[22,118], SSC[119], T1D[22,120], and VIT[121]. We also obtained GWAS data from FinnGen Release 5 website [https://www.finngen.fi/en/access_results], Pan-UK Biobank website [https://pan.ukbb.broadinstitute.org], and BioBank Japan PheWeb [https://pheweb.jp] for traits available in these biobanks. A more detailed information of each study can be found in Supplementary Data 1. For TWAS results, we provided TWAS association statistics from two distinct tissues, including DGN (whole blood) and GEUVADIS (lymphoblastoid cell line). We have also linked GWAS variant to its target gene by labeling the eQTL SNP in the gene expression prediction model with the smallest GWAS P value ("top variant" column). DGN data can be requested at https://www.nimhgenetics.org/request-access/how-to-request-access under "Depression Genes and Networks study (D. Levinson, PI)". GEUVADIS data can be accessed at https://www.ebi.ac.uk/arrayexpress/experiments/E-GEUV-1. Gene expression prediction models were created using PUMICE[41] and TWAS association statistics were calculated by TESLA[42]. DICE dataset can be requested through dbGaP accession number phs001703.v1.p1. B cell dataset from SLE subjects is available from the NCBI Gene Expression Omnibus under accession number GSE118256.

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

## Acknowledgements

This work was supported by the National Institutes of Health grants R01HG008983, R56HG011035, R01HG011035, R01GM126479, R21AI160138, R03OD032630, R01AI174108, R56HG012358, T32GM118294, T32LM012415, and U01AR071077. This work was also funded in part by the Penn State College of Medicine's Artificial Intelligence and Biomedical Informatics (AIBI) Program in the Strategic Plan, the Lupus Research Alliance, CURE funds from the Pennsylvania Department of Health, and by generous support from Robert and Sevia Finkelstein. BioVU acknowledgment: The datasets used for part of the PRS analysis were obtained from Vanderbilt University Medical Center's BioVU, which is supported by numerous sources: institutional funding, private agencies, and federal grants. These include the NIH funded Shared Instrumentation Grant S10OD017985 and S10RR025141; and CTSA grants UL1TR002243, UL1TR000445, and UL1RR024975. Its contents are solely the responsibility of the authors and do not necessarily represent official views of the National Center for Advancing Translational Sciences or the National Institutes of Health. Genomic data are also supported by investigator-led projects that include U01HG004798, R01NS032830, RC2GM092618, P50GM115305, U01HG006378, U19HL065962, R01HD074711; and additional funding sources listed at https://victr.vumc.org/biovu-funding/.

## Author contributions

C.K., L.T., B.J., and D.L. conceived the study. C.K. led the data analysis. C.K., Q.L., H.M., M.P., R.S., D.M., X.W., C.W., L.W., and S.C. conducted analyses. R.S., D.M., G.S., B.L., X.Z., N.O., and L.C. helped with data interpretation. C.K. and D.L. prepared the manuscript. All authors contributed to manuscript editing and approved the manuscript.

## Competing interests

The authors declare no competing interests.
