## [Peer Review File · Nature Communications]

Multi-ancestry and multi-trait genome-wide association meta-analyses inform clinical risk prediction for systemic lupus erythematosusREVIEWER COMMENTS

Reviewer #1 (Remarks to the Author):

This study presents a high quality metanalysis of SLE risk loci, connects risk loci to genes through several orthogonal assessments, nominates cell types contributing to the genetic risk, and identifies novel therapeutic opportunities based upon the genetic risk loci. Then they apply the genetic risk loci to the development of a polygenic risk score to assess the clinical utility of the identified genetic risk loci. The manuscript is clear and presents important data. It will be a strong resource for numerous groups studying SLE and SLE genetics.

1. This manuscript is written from the perspective of using genetic association data to inform the development of drugs and clinical treatment of patients. While this might well be a possibility in the future, the authors need to add a realistic context in the Discussion about what is needed to actually use PRS in a primary care (e.g., birth – line 98) setting. For example, in some of the presentations of effect, the PRS increases risk of an extremely rare disease (1:1000) by 2.5-fold – do the authors truly contend that this should be applied to people with no symptoms?
2. The rationale provided on line 76 for the TWAS was that it was important to link SLE-risk loci with non-coding variants to target genes. TWAS was then reported (line 78) to identify new SLE risk loci. It is important to also report the target genes linked to the SLE risk loci.
3. For the PRS assessment, it is important to clarify the discovery analysis used to identify genetic risk loci and the independent sets of subjects with and without SLE (that were not in the cohort used to identify genetic risk variants for SLE) that were used to train and then replicate the utility of the PRS. Please confirm explicitly that subjects in the PRS training/replication cohort were not in the GWAS studies used to derive the tested risk loci. When assessing the utility of the PRS, R² is insufficient. Please report: AUC, OR per SD, Pseudo R², OR of Top 5% vs bottom 5% for the training cohort and replication cohort. Importantly (and related to #1 above), the authors also need to provide the sensitivity and specificity of the PRS assessment and provide the false positive and true positive, false negative and true negative for the top 5% (or top 20%, if they choose).
4. The rationale for using 13 other autoimmune disorders to identify genetic risk for SLE is inadequately presented. If the goal of this study is to refine treatment of SLE specifically, it seems that it would be critical to be rigorous in the phenotype assessment of SLE. If the goal is to identify pan-autoimmune risk loci, then including other autoimmune cohorts could be appropriate.

Minor

1. In identifying the number of SLE risk loci, the authors should count not just loci that separated by chromosomal position, but should account for overlapping loci that are not in linkage disequilibrium and represent independent haplotypes that increase risk of SLE. A cut off of 0.2 r² within ancestries is standard for this type of LD pruning.
2. One line 134, the authors report not finding ancestry-specific novel loci using RANTES. Is this because there were fewer association studies to be consistent across for the specific ancestries or because RANTES was assessing consistency across all studies (not considering the ancestry of each study), and thus not identifying ancestry-specific loci?
3. One line 145, the authors report that 95% of sentinel (tag) variants were non-coding. What number of haplotypes (r²>0.8) at SLE loci contain coding variants? Note that previous work (including functional studies for SLE risk loci) have demonstrated that SLE risk loci with coding variants can also have genotype-dependent transcriptional regulatory activity based on the non-coding variants. This question is not intended to detract from the critical importance of non-coding variants and the important work of linking non-coding variants to genes.
4. SLE risk loci have been widely assessed for over a decade with much mechanistic work – applying all the loci to the “Open Target Gene” analysis described in line 145 would allow the authors to benchmark the results of loci with experimentally determined gene targets to support the accuracy of targets of novel loci. Also, the TWAS results in the shiny app are connected to GWAS tag variants – which is a missed opportunity to connect variants with genes.

5. The sentence on line 250 needs a strong reference to support the statement that SLE has a “lower proportion of causal variants across the genome compared to highly polygenic traits, such as height or smoking initiation.”

6. What was the criteria for a “positive” ANA or anti-dsDNA antibody test? Many people have “positive” tests, which patients with lupus have extremely high titers. Additionally, patients with SLE generally develop autoantibodies within 5-10 years of developing SLE. Using an electronic database, how are the authors approaching multiple tests of individual patients before and after developing SLE in this analysis?

7. The data availability URL and Shiny app could be useful, but more information is needed in the app to clarify which analysis from the manuscript is presented, which cohorts contributed to the analyses, and what methods were used.

Leah Kottyan

Reviewer #2 (Remarks to the Author):

Khunsriraksakul et al conducted a meta-analysis using publicly available multi-ancestry SLE GWAS datasets and multi-traits GWAS datasets on a number of autoimmune diseases to look for novel associations for SLE. A series of analyses were also conducted, including TWAS, drug repurposing based on genetics and gene expression data, and risk prediction using polygenic risk scores. Overall, this is a well-designed study, whose results add value to our understanding of this prototype autoimmune disease.

I have the following comments on the manuscript.

Many of the details are not given in the manuscript (preferably through supplementary information). For example, we don't have a clear idea that for a given novel locus, how much support is from SLE GWAS and how much support is from other autoimmune diseases. Instead, we have a statistical consistency value and much of the details are hidden under the statistical test. It should have given information on the support from SLE GWAS and support from other autoimmune diseases separately.

It was mentioned that all the novel loci are ancestry-shared. On the one hand, this is not surprising as the study design is biased towards finding the shared loci rather than the ancestry-specific loci. On the other hand, this is an over-simplification as both the strength of evidence and effect size heterogeneity are not adequately addressed and presented in the manuscript (including the supplementary).

As far as I know, the samples from EUR-1 (pmid: 26502338) have some overlap with that from EUR-2 (pmid: 28714469). Did the authors remove the overlapped samples?

A minor point: CLNK and RGS1 loci are known.

In general, the manuscript gives an impression that the study is aimed at breadth rather than depth. It might benefit from a more focused reconstruction. For example, the meta-analyses and GWAS and TWAS do not have equal values. This reviewer, I believe I am not alone, understands and has more confidence in the GWAS meta-analysis results, but is unsure how much we should trust the TWAS findings. The authors could address this issue by providing more detailed information on the TWAS results, especially the results on the known loci. This will help people to gauge the value of the novel TWAS findings. I believe overall, the genetics field has not broadly embraced the findings from TWAS, although there should be values in the analysis.

The introduction reads like an extended abstract.

I have a generally very positive view of the study. However, I also would like to point out some loose connections. For example, the PRS results are good on the surface, but this does not change the fact

that the authors do not have access to raw data of any case control cohort and have to rely on biobank data. The diagnosis of the biobank data is based on criteria that are different from other studies, which adds a layer of uncertainty to the results. I would like to hear the authors addressing this issue in a revision.

Using PRS in aiding SLE diagnosis indeed holds great potential. It is good that the authors extended the study to subjects with anti-ANA and anti-ds-DNA autoantibody information. The clinical utility of PRS might be better addressed by the added predictive value for these individuals. Age, sex, and ethnicity should be factored into the prediction. I personally believe that the comparison should be always between top and bottom quintile, which will make the results easier to follow and also make them a better reflection of the value of PRS. Eventually, having a PRS close to average of the general population might offer little value to the risk estimation for that individual, and it is the two extremes that will offer extra values.

In Figure 5, it is clear that MGI and BioVu datasets have differences, which is a reflection of a limitation of the study that should be addressed in the discussion. I believe the analysis of Figure 5 has too many comparisons. It should be focusing on comparisons between with or without PRS. In the line 251-252, the authors claimed that LASSOSUM risk scores have generated AUCs of 0.75 and 0.74 in BioVU and MGI, respectively. However, in the figure 5B, the AUCs for PRS in BioVU and MGI were reduced to around 0.68 and 0.65, respectively. Did the authors use different criteria to define SLE cases? or using a different model to calculate the PRS? The authors need to further explain the different results in the manuscript.

REVIEWER COMMENTS

Reviewer #1 (Remarks to the Author):

This study presents a high quality meta-analysis of SLE risk loci, connects risk loci to genes through several orthogonal assessments, nominates cell types contributing to the genetic risk, and identifies novel therapeutic opportunities based upon the genetic risk loci. Then they apply the genetic risk loci to the development of a polygenic risk score to assess the clinical utility of the identified genetic risk loci. The manuscript is clear and presents important data. It will be a strong resource for numerous groups studying SLE and SLE genetics.

1. This manuscript is written from the perspective of using genetic association data to inform the development of drugs and clinical treatment of patients. While this might well be a possibility in the future, the authors need to add a realistic context in the Discussion about what is needed to actually use PRS in a primary care (e.g., birth – line 98) setting. For example, in some of the presentations of effect, the PRS increases risk of an extremely rare disease (1:1000) by 2.5-fold – do the authors truly contend that this should be applied to people with no symptoms?

Response: Thank you for the comment! This is an important point. We have included the following text in the Discussion section. “Our work showed that PRS when used in conjunction with conventional lab values such as ANA and anti-dsDNA levels, could improve the diagnostic accuracy. PRS would not replace conventional lab tests, but as germline DNA usually would not change over lifetime, the use of PRS could facilitate early diagnosis and risk screening. PRS has the most clinical utility in individuals with extreme PRS values, who will benefit from careful monitoring for progression⁸⁶. With the improvement in the understanding of polygenic architecture of complex diseases, the advancement of statistical methodology, and the increase of genetic diversity in genotyped samples, we could expect further improvements in the accuracy of genetic prediction⁸⁶” [Lines 368-376].

2. The rationale provided on line 76 for the TWAS was that it was important to link SLE-risk loci with non-coding variants to target genes. TWAS was then reported (line 78) to identify new SLE risk loci. It is important to also report the target genes linked to the SLE risk loci.

Response: We have provided this information in **Supplementary Tables 7-8**.

3. For the PRS assessment, it is important to clarify the discovery analysis used to identify genetic risk loci and the independent sets of subjects with and without SLE (that were not in the cohort used to identify genetic risk variants for SLE) that were used to train and then replicate the utility of the PRS. Please confirm explicitly that subjects in the PRS training/replication cohort were not in the GWAS studies used to derive the tested risk loci. When assessing the utility of the PRS, R² is insufficient. Please report: AUC, OR per SD, Pseudo R², OR of Top 5% vs bottom 5% for the training cohort and replication cohort. Importantly (and related to #1 above), the authors also need to provide the sensitivity and specificity of the PRS assessment and provide the false

positive and true positive, false negative and true negative for the top 5% (or top 20%, if they choose).

Response: We have now clarified in the main text that our testing cohorts (i.e., MGI and BioVU datasets) were not part of the discovery analysis used to create GWAS summary statistics and train polygenic risk scores [Line 270]. Importantly, we used summary-based PRS methods in all of our PRS analyses, which do not require hyperparameter tuning and hence do not need a validation dataset. We have reported AUC, R^2 on a liability scale, true negative, true positive, false negative, false positive, sensitivity, specificity, odd ratio (OR) per standard deviation, OR of top 20% vs. bottom 20%, and Nagelkerke's R^2 in **Supplementary Figure 6** and **Supplementary Table 9** [Lines 272-276].

4. The rationale for using 13 other autoimmune disorders to identify genetic risk for SLE is inadequately presented. If the goal of this study is to refine treatment of SLE specifically, it seems that it would be critical to be rigorous in the phenotype assessment of SLE. If the goal is to identify pan-autoimmune risk loci, then including other autoimmune cohorts could be appropriate.

Response: Thank you for the comment! Studies have shown that autoimmune diseases have pervasively shared genetic basis. Exploiting the shared genetic effects and jointly analyzing multiple traits will improve power for loci with pleiotropic effects, as traits with smaller sample sizes will borrow strength from traits with larger sample sizes. Based on this idea, we use MTAG to analyze correlated autoimmune diseases. MTAG gave unbiased genetic effect estimates for each trait, which means that identified loci are associated with SLE, not just a pan-disease loci. As we showed in **Supplementary Table 4**, the SLE-only multi-ancestry meta-analysis results using only SLE summary statistics show significant or border-line significant p-values for loci identified by MTAG. Results from MTAG will also improve the accuracy of genetic risk scores.

Minor

1. In identifying the number of SLE risk loci, the authors should count not just loci that separated by chromosomal position, but should account for overlapping loci that are not in linkage disequilibrium and represent independent haplotypes that increase risk of SLE. A cut off of 0.2 r^2 within ancestries is standard for this type of LD pruning.

Response: Thank you for the suggestion. We created a reference panel with the same ancestry composition as the SLE dataset using the 1000 Genomes Project. We then performed LD pruning using PLINK with the created reference panel and an r^2 cut-off of 0.2. In total, we found 249 loci, in comparison to 106 loci defined according to 1 million basepair window surrounding the sentinel variants [Lines 122-128].

2. One line 134, the authors report not finding ancestry-specific novel loci using RATES. Is this because there were fewer association studies to be consistent across for the specific ancestries or because RATES was assessing consistency across all studies (not considering the ancestry of each study), and thus not identifying ancestry-specific loci?

Response: Thank you for the question! Our results are similar to the studies of other traits [<https://www.medrxiv.org/content/10.1101/2021.11.19.21266436v1>], where a majority of loci (96%) have homogeneous effects across ancestries. RATES extends MAMBA and already takes ancestral heterogeneity into account. It uses meta-regression to adjust for ancestral heterogeneity, and then assess if the residuals after the adjustment show consistent signals across different studies. We do not use RATES for association analysis and discovery, but instead use it to assess if the identified association signal is genuine and may be replicable if a sufficiently large replication dataset is available. In fact, we plotted the effect sizes of the non-replicable variants across studies. As shown in **Supplementary Figure 4B**, these non-replicable variants often show unusually large effect sizes disproportional to their sample size (e.g., rs10101368), which indicates that they are spurious association signals.

3. One line 145, the authors report that 95% of sentinel (tag) variants were non-coding. What number of haplotypes ($r^2 > 0.8$) at SLE loci contain coding variants? Note that previous work (including functional studies for SLE risk loci) have demonstrated that SLE risk loci with coding variants can also have genotype-dependent transcriptional regulatory activity based on the non-coding variants. This question is not intended to detract from the critical importance of non-coding variants and the important work of linking non-coding variants to genes.

Response: Thank you for the suggestion. Out of 106 SLE risk loci (defined by chromosomal position), we found 22 loci (20.8%) to contain sentinel/tagged variants ($r^2 > 0.8$) in the coding regions [Lines 155-156]. Out of 249 SLE risk loci (defined by LD pruning), we found 36 loci (14.5%) to contain sentinel/tagged variants ($r^2 > 0.8$) in coding regions.

4. SLE risk loci have been widely assessed for over a decade with much mechanistic work – applying all the loci to the “Open Target Gene” analysis described in line 145 would allow the authors to benchmark the results of loci with experimentally determined gene targets to support the accuracy of targets of novel loci. Also, the TWAS results in the shiny app are connected to GWAS tag variants – which is a missed opportunity to connect variants with genes.

Response: Thank you for the suggestion. We have incorporated the information from Lu et al. (PMID: 33712590) by linking variants with enhancer activity (enVars) to its target genes using Open Target Genetics database. In total, we were able to link these enVars to 26 unique target genes and subsequently used these genes as a reference to benchmark the accuracy of our TWAS results. We found 22 enVar target genes overlapping TWAS genes with nominal significance levels, and 9 enVar target genes overlapping TWAS hits with significant P values under the Bonferroni threshold for testing multiple genes genome-wide [Lines 181-187]. We have also linked GWAS variant to its target gene by labeling the eQTL SNP in the gene expression prediction model with the smallest GWAS P value. We provide this information on the Shiny app (“top variant” column) [Lines 408-410].

5. The sentence on line 250 needs a strong reference to support the statement that SLE has a “lower

proportion of causal variants across the genome compared to highly polygenic traits, such as height or smoking initiation.”

Response: Thank you for the suggestion and sorry for the confusion! We observed that polygenic risk score methods that impose sparsity tend to outperform methods that do not impose sparsity, which may indicate that the genetic architecture for SLE tends to be more sparse than highly polygenic traits. Due to sample sizes differences between different traits and limited statistical power, the fraction of causal variants is often very difficult to estimate and compare. We have thus removed the sentence from the revised manuscript.

6. What was the criteria for a “positive” ANA or anti-dsDNA antibody test? Many people have “positive” tests, which patients with lupus have extremely high titers. Additionally, patients with SLE generally develop autoantibodies within 5-10 years of developing SLE. Using an electronic database, how are the authors approaching multiple tests of individual patients before and after developing SLE in this analysis?

Response: This information can be found in **Methods** section. For ANA test result, we considered a titer of $\geq 1:80$ (e.g., 1:80, 1:160, 1:320, etc.) as positive, and the other values as negative (e.g., negative status, 1:40, 1:20, and etc.). For anti-dsDNA test result, we considered a level of ≥ 60 IU/mL or positive status as positive, and the other values as negative (e.g., < 60 IU/mL or negative status). Individuals with any positive test results are considered positive [Line 594].

7. The data availability URL and Shiny app could be useful, but more information is needed in the app to clarify which analysis from the manuscript is presented, which cohorts contributed to the analyses, and what methods were used.

Response: We have clarified the Data Availability statement to reflect the details of the analyses being performed. “For GWAS result, we provided the summary statistics information of the multi-ancestry and multi-trait SLE meta-analysis result. This meta-analysis result was derived via MTAG³³ and METAL⁸⁸ from the following datasets: SLE^{14,19-26,89}, ATD²⁵, CD^{90,91}, CEL^{92,93}, MS⁹³⁻⁹⁵, PBC^{96,97}, RA^{25,98}, SJO^{25,99}, SSC¹⁰⁰, T1D^{25,101}, and VIT¹⁰². When available, we also obtained GWAS data from FinnGen website (Release 5 data; https://www.finnngen.fi/en/access_results), Pan-UK Biobank website (<https://pan.ukbb.broadinstitute.org/>), and BioBank Japan PheWeb (<https://pheweb.jp>). A more detailed information of each study can be found in **Supplementary Table 1**. For TWAS results, we provided TWAS association statistics from two distinct tissues, including DGN (whole blood) and GEUVADIS (lymphoblastoid cell line). We have also linked GWAS variant to its target gene by labeling the eQTL SNP in the gene expression prediction model with the smallest GWAS P value (“top variant” column). DGN data can be requested at <https://www.nimhgenetics.org> under “Depression Genes and Networks study (D. Levinson, PI)”. GEUVADIS data can be accessed at <https://www.ebi.ac.uk/arrayexpress/experiments/E-GEUV-1>. Gene expression prediction models were created using PUMICE⁴⁴ and TWAS association statistics were calculated by TESLA⁴⁵.” [Lines 399-414].

Leah Kottyan

Reviewer #2 (Remarks to the Author):

Khunsriraksakul et al conducted a meta-analysis using publicly available multi-ancestry SLE GWAS datasets and multi-traits GWAS datasets on a number of autoimmune diseases to look for novel associations for SLE. A series of analyses were also conducted, including TWAS, drug repurposing based on genetics and gene expression data, and risk prediction using polygenic risk scores. Overall, this is a well-designed study, whose results add value to our understanding of this prototype autoimmune disease.

I have the following comments on the manuscript.

Many of the details are not given in the manuscript (preferably through supplementary information). For example, we don't have a clear idea that for a given novel locus, how much support is from SLE GWAS and how much support is from other autoimmune diseases. Instead, we have a statistical consistency value and much of the details are hidden under the statistical test. It should have given information on the support from SLE GWAS and support from other autoimmune diseases separately.

Response: We have added the additional information in **Supplementary Table 4**. "Across 16 novel and replicable loci, we found that multi-ancestry SLE-only GWAS already yielded genome-wide significant p-values in 3 loci and borderline significant P values for the remaining 13 loci (highest P value = 8.19×10^{-5}). Importantly, rheumatoid arthritis's GWAS contributes the most to the identification of novel loci in SLE, having the smallest P values for 8 out of 16 loci in comparison to other autoimmune diseases (**Supplementary Table 4**). This is likely due to the largest sample size of autoimmune GWAS data and the overlap of clinical features between SLE and RA³⁵" [Lines 134-140].

rsID	Variant ID	SLE	ATD	CD	CEL	MS	PBC	RA	SJO	SSC	T1D	UC
rs662618	1_92935411_T_C_b37	1.48 $\times 10^{-10}$	9.27 $\times 10^{-1}$	2.43 $\times 10^{-3}$	8.21 $\times 10^{-1}$	8.39 $\times 10^{-7}$	9.31 $\times 10^{-1}$	5.48 $\times 10^{-5}$	2.01 $\times 10^{-2}$	3.07 $\times 10^{-2}$	2.91 $\times 10^{-3}$	1.64 $\times 10^{-1}$
rs2453044	1_120508524_A_G_b37	4.10 $\times 10^{-6}$	4.49 $\times 10^{-1}$	1.03 $\times 10^{-4}$	8.53 $\times 10^{-1}$	8.94 $\times 10^{-1}$	6.17 $\times 10^{-2}$	1.55 $\times 10^{-1}$	3.69 $\times 10^{-1}$	9.58 $\times 10^{-1}$	4.94 $\times 10^{-7}$	6.12 $\times 10^{-1}$
rs12992553	2_70360262_A_G_b37	4.59 $\times 10^{-6}$	7.35 $\times 10^{-1}$	6.65 $\times 10^{-1}$	2.34 $\times 10^{-1}$	5.13 $\times 10^{-1}$	1.43 $\times 10^{-2}$	1.01 $\times 10^{-2}$	2.85 $\times 10^{-1}$	1.14 $\times 10^{-3}$	1.38 $\times 10^{-1}$	9.64 $\times 10^{-1}$
rs13014122	2_135050622_G_A_b37	2.03 $\times 10^{-8}$	4.28 $\times 10^{-1}$	5.17 $\times 10^{-1}$	4.27 $\times 10^{-1}$	1.81 $\times 10^{-1}$	4.18 $\times 10^{-2}$	2.53 $\times 10^{-2}$	9.03 $\times 10^{-1}$	1.38 $\times 10^{-1}$	3.31 $\times 10^{-1}$	9.03 $\times 10^{-1}$
rs299629	3_12576846_A_G_b37	6.08 $\times 10^{-5}$	4.41 $\times 10^{-1}$	9.56 $\times 10^{-3}$	9.11 $\times 10^{-1}$	2.91 $\times 10^{-1}$	2.18 $\times 10^{-2}$	7.30 $\times 10^{-5}$	4.05 $\times 10^{-2}$	4.92 $\times 10^{-4}$	8.56 $\times 10^{-3}$	9.71 $\times 10^{-1}$
rs12490565	3_121553719_G_A_b37	1.76 $\times 10^{-7}$	7.40 $\times 10^{-1}$	2.47 $\times 10^{-1}$	2.96 $\times 10^{-1}$	6.35 $\times 10^{-2}$	1.77 $\times 10^{-2}$	1.20 $\times 10^{-3}$	6.40 $\times 10^{-1}$	2.05 $\times 10^{-3}$	2.34 $\times 10^{-1}$	7.25 $\times 10^{-2}$
rs4697651	4_10721433_C_T_b37	4.01 $\times 10^{-6}$	1.14 $\times 10^{-2}$	2.03 $\times 10^{-1}$	1.64 $\times 10^{-1}$	2.62 $\times 10^{-1}$	7.37 $\times 10^{-1}$	1.46 $\times 10^{-5}$	8.02 $\times 10^{-1}$	7.68 $\times 10^{-1}$	1.67 $\times 10^{-2}$	4.24 $\times 10^{-1}$
rs2288786	5_102600754_G_A_b37	8.19 $\times 10^{-5}$	3.62 $\times 10^{-1}$	1.45 $\times 10^{-2}$	1.54 $\times 10^{-1}$	2.51 $\times 10^{-1}$	1.27 $\times 10^{-7}$	1.66 $\times 10^{-8}$	4.97 $\times 10^{-2}$	6.73 $\times 10^{-1}$	1.48 $\times 10^{-2}$	1.59 $\times 10^{-1}$
rs12529514	6_14096658_T_C_b37	1.70 $\times 10^{-5}$	9.32 $\times 10^{-2}$	1.85 $\times 10^{-1}$	9.06 $\times 10^{-1}$	5.85 $\times 10^{-1}$	3.46 $\times 10^{-2}$	4.52 $\times 10^{-11}$	2.21 $\times 10^{-1}$	2.66 $\times 10^{-1}$	1.62 $\times 10^{-1}$	8.28 $\times 10^{-1}$
rs6939565	6_130194204_C_T_b37	5.38 $\times 10^{-5}$	4.13 $\times 10^{-1}$	1.05 $\times 10^{-1}$	9.39 $\times 10^{-1}$	4.22 $\times 10^{-1}$	4.22 $\times 10^{-4}$	6.75 $\times 10^{-3}$	5.19 $\times 10^{-2}$	1.57 $\times 10^{-1}$	1.95 $\times 10^{-3}$	4.15 $\times 10^{-2}$
rs9494331	6_136006301_G_A_b37	1.49 $\times 10^{-6}$	1.36 $\times 10^{-1}$	6.23 $\times 10^{-1}$	6.53 $\times 10^{-1}$	5.12 $\times 10^{-2}$	2.66 $\times 10^{-2}$	2.33 $\times 10^{-1}$	1.50 $\times 10^{-1}$	3.07 $\times 10^{-1}$	6.37 $\times 10^{-3}$	1.73 $\times 10^{-1}$
rs3761847	9_123690239_G_A_b37	2.32 $\times 10^{-6}$	1.72 $\times 10^{-1}$	7.23 $\times 10^{-1}$	4.16 $\times 10^{-3}$	9.11 $\times 10^{-1}$	4.08 $\times 10^{-1}$	3.59 $\times 10^{-8}$	6.88 $\times 10^{-5}$	4.10 $\times 10^{-2}$	2.39 $\times 10^{-3}$	2.97 $\times 10^{-1}$
rs6602588	10_12487996_G_A_b37	4.35 $\times 10^{-8}$	2.71 $\times 10^{-2}$	1.15 $\times 10^{-1}$	1.25 $\times 10^{-1}$	3.14 $\times 10^{-1}$	4.22 $\times 10^{-1}$	9.75 $\times 10^{-1}$	5.81 $\times 10^{-1}$	4.99 $\times 10^{-3}$	2.30 $\times 10^{-1}$	7.77 $\times 10^{-1}$
rs516124	11_64128423_G_T_b37	1.46 $\times 10^{-6}$	6.06 $\times 10^{-2}$	1.48 $\times 10^{-5}$	4.34 $\times 10^{-1}$	7.11 $\times 10^{-1}$	3.75 $\times 10^{-10}$	3.38 $\times 10^{-10}$	2.51 $\times 10^{-1}$	1.19 $\times 10^{-2}$	1.27 $\times 10^{-5}$	7.38 $\times 10^{-1}$
rs195933	17_44828931_G_A_b37	3.28 $\times 10^{-6}$	7.80 $\times 10^{-1}$	3.88 $\times 10^{-2}$	4.27 $\times 10^{-1}$	9.35 $\times 10^{-1}$	2.20 $\times 10^{-4}$	7.12 $\times 10^{-1}$	6.38 $\times 10^{-4}$	3.08 $\times 10^{-2}$	1.70 $\times 10^{-5}$	4.97 $\times 10^{-1}$
rs1535271	20_57734753_G_A_b37	9.14 $\times 10^{-7}$	7.22 $\times 10^{-1}$	2.91 $\times 10^{-2}$	7.72 $\times 10^{-3}$	6.09 $\times 10^{-1}$	4.69 $\times 10^{-5}$	2.46 $\times 10^{-3}$	1.23 $\times 10^{-1}$	1.30 $\times 10^{-1}$	9.09 $\times 10^{-2}$	3.31 $\times 10^{-3}$

Disease name abbreviations: AN = ankylosing spondylitis, ATD = autoimmune thyroid disease, CD = Crohn's disease, CEL = celiac disease, MS = multiple sclerosis, PBC = primary biliary cirrhosis, PSOA = psoriatic arthritis, RA = rheumatoid arthritis, SJO = Sjogren's syndrome, SLE = systemic lupus erythematosus, SSC = systemic sclerosis, T1D = type 1 diabetes, UC = ulcerative colitis, VIT = vitiligo.

It was mentioned that all the novel loci are ancestry-shared. On the one hand, this is not surprising

as the study design is biased towards finding the shared loci rather than the ancestry-specific loci. On the other hand, this is an over-simplification as both the strength of evidence and effect size heterogeneity are not adequately addressed and presented in the manuscript (including the supplementary).

Response: Thank you for the comment! To investigate the extent of genetic effect heterogeneity, we conducted the Cochran's Q test for heterogeneity for 106 loci. We found that 97% of the loci did not show significant p-values for the test of heterogeneity after correcting for multiple comparisons (i.e., P values for Cochran's Q test $\geq 0.05/106$) (**Supplementary Table 5**). As fixed effect meta-analysis favors loci with homogeneous effects, we further explored the extent of heterogeneity using additional sub-threshold variants ($P < 1 \times 10^{-6}$), and still found that 97% of 144 loci did not show evidence of heterogeneity in effect sizes across different SLE studies with P values for Cochran's Q test $\geq 0.05/144$ [Lines 142-145]. This finding is in agreement with the recent observation from the Global Biobank Meta-analysis Initiative where the authors found that ~96% of the loci did not show evidence in heterogeneity in effect sizes across different datasets (<https://www.medrxiv.org/content/10.1101/2021.11.19.21266436v1>).

As far as I know, the samples from EUR-1 (pmid: 26502338) have some overlap with that from EUR-2 (pmid: 28714469). Did the authors remove the overlapped samples?

Response: Thank you for pointing this out. In the previous analysis, we systematically examined potential sample overlaps using the λ_{meta} statistic. The two cohorts did not stand out as overlapping (i.e., $\lambda_{meta} = 1.08$ in **Supplementary Figure 8**). To adjust for the potential correlated statistics between these two studies, we have now followed a general framework outlined in PMID: 26908615 for meta-analyzing studies with overlapping subjects. The method works by decorrelating the summary statistics from overlapping studies and combining the decorrelated summary statistics using standard meta-analysis methods. We found that the multi-ancestry GWAS meta-analysis results with/without overlapped sample adjustments showed nearly identical results (Figure below) and our conclusions remain.

A minor point: *CLNK* and *RGS1* loci are known.

Response: Thank you for the comment! We noted that the authors of a recent SLE GWAS article (PMID: 33536424) included these two loci. For *CLNK* locus, the P values of rs4697651 in EAS, EUR, and EAS+EUR populations are 4.49×10^{-5} , 3.26×10^{-4} , and 6.36×10^{-8} , none of which reaches the P value threshold for genome-wide significance (5×10^{-8}). We therefore consider *CLNK* locus as novel in our study. For *RGS1* locus, the P values of rs1547624 in EAS, EUR, and EAS+EUR populations are 4.55×10^{-8} , 1.38×10^{-1} , and 1.95×10^{-7} and the reported p-value in the GWAS catalog was rounded to 5×10^{-8} . We have now considered *RGS1* locus as a known locus in the revised manuscript.

In general, the manuscript gives an expression that the study is aimed at breadth rather than depth. It might benefit from a more focused reconstruction. For example, the meta-analyses and GWAS and TWAS do not have equal values. This reviewer, I believe I am not alone, understands and has more confidence in the GWAS meta-analysis results, but is unsure how much we should trust the TWAS findings. The authors could address this issue by providing more detailed information on the TWAS results, especially the results on the known loci. This will help people to gauge the value of the novel TWAS findings. I believe overall, the genetics field has not broadly embraced the findings from TWAS, although there should be values in the analysis.

Response: Thank you for the suggestion. We have incorporated the information from Lu et al. (PMID: 33712590) by linking enhancer variants (enVars) to its target genes using Open Target

Genetics. In total, we were able to link these enVars to 26 unique target genes and subsequently used these genes as a reference to benchmark the accuracy of our TWAS results. We found 22 enVar target genes overlapping TWAS genes with nominal significance levels, and 9 enVar target genes overlapping TWAS hits with significant P values under the Bonferroni threshold for testing multiple genes genome-wide [Lines 181-187]. We have also provided the TWAS hits at the 106 GWAS loci in **Supplementary Tables 7-8**. We hope that these results may help readers better interpret TWAS results.

The introduction reads like an extended abstract.

Response: We have revised the **Introduction** section to provide more background information and motivation for the manuscript [Lines 56-90].

I have a generally very positive view of the study. However, I also would like to point out some loose connections. For example, the PRS results are good on the surface, but this does not change the fact that the authors do not have access to raw data of any case control cohort and have to rely on biobank data. The diagnosis of the biobank data is based on criteria that are different from other studies, which adds a layer of uncertainty to the results. I would like to hear the authors addressing this issue in a revision.

Response: We have performed GWAS meta-analysis on MGI and BioVU datasets and compared the effect sizes to those from the multi-ancestry and multi-trait meta-analysis GWAS result (**Supplementary Figure 5**). Below, we made scatter plot to compare the effect sizes (beta) from multi-ancestry and multi-trait meta-analysis (X-axis) and the effect sizes (beta) from MGI and BioVU meta-analysis (Y-axis). Here, we only showed variants reaching genome-wide significance (p value $< 5e-8$) in the multi-ancestry and multi-trait meta-analysis. Blue dots represent significant variants in MGI and BioVU meta-analysis result. Despite the small number of cases in BioVU and MGI datasets, we observed a good concordance between the two GWAS results (Pearson correlation coefficient of 0.7 and p -value = $4.4e-16$) (Figure below).

Using PRS in aiding SLE diagnosis indeed holds great potential. It is good that the authors extended the study to subjects with anti-ANA and anti-ds-DNA autoantibody information. The clinical utility of PRS might be better addressed by the added predictive value for these individuals. Age, sex, and ethnicity should be factored into the prediction. I personally believe that the comparison should be always between top and bottom quintile, which will make the results easier to follow and also make them a better reflection of the value of PRS. Eventually, having a PRS close to average of the general population might offer little value to the risk estimation for that individual, and it is the two extremes that will offer extra values.

Response: Thank you for the suggestion! All prediction performances (e.g., Nagelkerke's R2 on the liability scale and AUC), were calculated after adjusting for sex, PC1-10, and HLA alleles. We have also added the following texts to further describe the added predictive value by PRS:

- “Specifically, in BioVU, the AUC of using PRS + ANA + anti-dsDNA is 0.75, which improves the AUC of using only ANA + anti-dsDNA (0.73) with P value 0.005 (**Figure 5B**). Similarly, in MGI, the AUC of using PRS + ANA + anti-dsDNA is 0.75, which improves the AUC of using ANA + anti-dsDNA alone (0.74) with a borderline significant P value 0.065. When jointly analyzing the MGI and BioVU cohorts, the AUC improves from 0.72 with ANA + anti-dsDNA alone to 0.74 with PRS + ANA + anti-dsDNA (P value = 0.002)” [Lines 321-326].
- “Among individuals that are ANA positive or anti-dsDNA-negative, the ones with PRS in the top 20th percentile are approximately 2.36 and 2.34 times more likely to develop SLE than those with PRS in the bottom 20th percentile, respectively (**Figure 5C**). When focusing on patients who had positive ANA and negative anti-dsDNA tests in the MGI and BioVU biobanks (the most common lab test results observed clinically), patients with PRS in the top 20th percentile are approximately 2.31 times more likely to have SLE than those with PRS in the bottom 20th percentile” [Lines 329-335].

We also agree with the reviewer to focus on individuals with extreme PRS values. We have modified our **Figure 5A** to reflect the comparison between the top 20% and bottom 20% groups.

In Figure 5, it is clear that MGI and BioVu datasets have differences, which is a reflection of a limitation of the study that should be addressed in the discussion. I believe the analysis of Figure 5 has too many comparisons. It should be focusing on comparisons between with or without PRS.

Response: We have addressed the limitation of the study in the **Discussion** section. “The results in MGI and BioVU datasets slightly differ, which may be due to factors such as geographic catchment and sampling strategy²⁷. Nonetheless, the results are, in general, concordant with each other and demonstrate the added utility of PRS in diagnosing SLE” (Line 358-362). We have also made changes to **Figure 5** to focus on comparisons between clinical lab tests with / without PRS.

In the line 251-252, the authors claimed that LASSOSUM risk scores have generated AUCs of 0.75 and 0.74 in BioVU and MGI, respectively. However, in the figure 5B, the AUCs for PRS in BioVU and MGI were reduced to around 0.68 and 0.65, respectively. Did the authors use different criteria

to define SLE cases? or using a different model to calculate the PRS? The authors need to further explain the different results in the manuscript.

Response: Thank you for pointing this out. Indeed, the analysis performed in **Figure 5B** looks at the subset of patients with both ANA and anti-dsDNA test results (MGI: $N_{\text{case}} = 282$, $N_{\text{control}} = 1,074$; BioVU: $N_{\text{case}} = 342$, $N_{\text{control}} = 1,408$). The other analysis, lines 251-252 of the originally submitted manuscript, uses all cases and controls in the dataset, regardless of whether they have lab value measurements. We have clarified this in **Figure 5B** and its legend.

REVIEWER COMMENTS

Reviewer #1 (Remarks to the Author):

The authors comprehensively addressed my major and minor critiques.

Reviewer #2 (Remarks to the Author):

The authors have conducted a multi-ethnicity, multi-trait meta-analysis for SLE, and presented results on TWAS, drug repurposing analysis and PRS. The study should add important information to the field of SLE genetics.

Multi-trait analyses is based on an assumption that different autoimmune diseases may share the same association signals. In general, sharing a locus among autoimmune diseases is a common phenomenon but sharing the same association signal is not systematically examined. The authors conducted LD score regression to evaluate distances among the diseases and should partially address the question. I have some concerns if the major signal comes from other autoimmune diseases rather than SLE. For example, *CCDC88B* is a well-known locus that is associated with MS, alopecia areata, autoimmune thyroid diseases, PBC and Crohn's disease. In this case, I believe there should be a high threshold on result from SLE multi-ethnicity analysis before multi-trait analysis is conducted. If the SLE signal is low (may not be the case), and if the major signal comes from other autoimmune diseases, the validity of a SLE novel association would be in question. I hope the authors adopt a stringent standard in this part of analysis and clearly state it.

The couple of percentage improvement in PRS does not convince me the application value of PRS in this scenario. In what situations that PRS would help a physician make a decision? I believe the best chances are for individuals with the highest and lowest PRS, the two ends of the spectrum, and in situations that a decision is hard to make, for example based on ANA and antids-DNA autoantibody alone. PRS scores will not help the individuals with PRS in the middle but if the authors can present a case that for those with extremely high or low PRS, this can clearly help with a clinical decision making, it will be great.

TWAS results and drug repurposing: the analyses clearly provide useful information, but how to gauge the strength of the evidence? Could the connection go beyond merely a suggestive possibility for an available drug as efficacious for SLE?

Minor points: line 189-196, the description of tissue specificity is little value comparing LCL and PBMC, I believe. Supplementary Table 6 may benefit the reader if it contains information on reported gene/closest gene(s).

Reviewer #3 (Remarks to the Author)

General comments: this is an important study, worthy of report in Nat Comm. I did not notice any major methodological flaws or other errors but the standard of write-up is not high enough. I think the manuscript would benefit from a thorough rewriting that eliminated excessive wording (I think about 25% could go), and improved the organisation. Particularly the sections in red (which appear to be edits from an earlier round of review in which I was not involved) are often verbose and not well integrated with the original text. To cite one very small issue, the authors switch between using "LD" and spelling out "linkage disequilibrium" breaking a basic rule, once you introduce an acronym (and LD was not defined) use it consistently. That's a minor point but symptomatic of insufficient care. Most important is that too much of the wording is vague so that the exact meaning is fudged – I have specific examples below. Datasets are one important area for improvement, they should be described

and named in one place and then referred to exactly by the specified name.

Some key parts of the analysis are not multi-ancestry, e.g. the selection of genetically-correlated traits. Because the title emphasises multi-ancestry, it should be more clear e.g. in a workflow diagram, what aspects of the analysis were not multi-ancestry.

Regarding the genetically-correlated traits, significant at $FDR < 0.05$ is a weak criterion, I would have preferred an additional threshold on the estimated correlation coefficient to make the analysis more specific to SLE. Otherwise traits poorly-correlated with SLE can be collectively more important than SLE in the MTAG analysis.

Even based on this very large dataset, prediction accuracy using PRS is not high (as for many other complex conditions): the AUC is modest and shows only minimal improvement over current lab tests. A more honest discussion needs to be had of its limited utility: I can't foresee any clinical use for prediction at the accuracy shown here. If the authors disagree they should argue the case not just make vague claims lacking justification such as L371 "the use of PRS could facilitate early diagnosis and risk screening".

Specific points:

L43 "16 novel GWAS loci and an additional 22 novel loci" I think these are different definitions of "loci" and it would be better to indicate this with different words.

L58 "ranges from 20 to 150" is there any explanation for such a wide range? Are these for different sectors of the general population, e.g. different ancestry groups?

L112 "significant SLE-correlated traits (false discovery rate < 0.05)" this text suggests phenotypic correlations but it becomes clear that the authors are talking about genetic correlations.

L113 "of significant genetic correlation with SLE" presumably this is referring to the 10 disorders in the previous sentence (now correctly described as genetically-correlated)? Much clearer to say that.

L123 "the linkage equilibrium-based pruning" better to call this "LD-based" as in L125

L126 typo: r^2

L130 I know this term comes from other authors but "replicability" should not be used without details/explanation. This term is misleading and I would prefer it was avoided, but at least say "this term was introduced by X to mean ...". Whether a hit is replicable depends on the details of the replication study, but most important is whether it is a valid association with effect size above some threshold. Instead "replicability" here is being used for a criterion of consistency of effect sizes implicitly assuming that consistency is an indicator of true effect – that needs an honest explanation, not just a misleading name.

L133 "posterior probability of replicability" this is even more confusing, it's not even grammatical: whatever "replicability" means, it is not a binary term that can have a posterior probability. A statement that would be meaningful is "posterior probability that the true effect size exceeds X ...". Again give some explanation of what this means, and it should be used in quotes to highlight that the term is misleading but attributed to other authors.

L141 "these novel loci are all shared across ancestries" state the criterion for "shared"

L148 "A portion of previously known loci" quantify this – how many are you aware of? "most of them are still marginally significant" quantify "most" and "marginally". Currently this claim is too vague to

be useful.

L155 "95% of identified sentinel variants are non-coding. However, it should be noted that 79% of sentinel variant and its associated tag variants are located in the non-coding region." I find this confusing – the new text needs to be better integrated with the original text.

L172 "7,028 and 9,260 significant gene expression prediction models" since a "model" is very general I don't know what is being counted here: are these numbers of significant genes? Also, significant compared with what?

L175 "optimally integrate eQTL data of European ancestry with a multi-ancestry GWAS dataset" The wording is too vague for me to understand: it's not enough to bury details in the methods, you have to allow readers of the main text to understand the key ideas. What is being optimised? What is meant by "integrate"? Are the "eQTL data of European ancestry" the datasets described above and similarly is "a multi-ancestry GWAS dataset" the same GWAS dataset described above?

L190 When reporting 40 and 38, remind us out of how many.

L248 "which serves as a positive control" if you want to claim this you need to describe the experimental design and analysis that justify it. Deleting this phrase would make the sentence just acceptable in my view, but strictly the "Some" at the start of the sentence is so weak that nothing follows from it.

L260 "1) a large European GWAS" GWAS for what trait? Of what size? If for SLE, you need to state its overlap with the GWAS described here. For 2) and 3) I assume the datasets are those described above in this manuscript, but why introduce new terminology here? (Also I think "MA & MT" may confuse and better to write MAMT). L263 introduces two datasets not in this list. I find the presentation to be confusing.

L269 What is a "model" here? Also, sample sizes for the datasets – and statement of their relationship to the primary dataset - should be given above, at first mention.

L278 "sparsity" how "sparse" is LASSOSUM? Also is it significantly better than other methods? Where are the reports of predictive accuracy and their SEs for all the PRS methods? You imply that LASSOSUM is the only sparse method, but pruning and thresholding is also sparse.

L281 "performed the best" best compared with what? Are you just saying that MAMT beats MA? If so the superlative "best" is misleading.

L285 "biobanks based on electronic medical records" As stated above, better to consolidate descriptions of datasets in one place not introduce in bits and pieces. Why is EMR data relevant to the decision to examine sensitivity?

L316 "more likely to develop SLE" I accept that such probability statements are widely used, but unnecessary and I think we'd be better off if authors made a more direct statement of facts: you observed/estimated/computed that there were 3.8 times as many cases in ... than in ...

L324 change "borderline significant" to "non-significant". It seems disappointing that PRS gives so little advantage over the biomarkers alone, since the latter were said to have "low accuracy". Also the biomarkers seem to add nothing to the AUC of PRS alone – this observation should be explicitly reported.

L490 "produce a larger number of significant gene expression models" similar to comment above, "models" is ambiguous here and I can't understand the claim.

Fig 1 (right) avoid using significance as a proxy for effect size as it can be misleading. Use significance as a threshold, but show estimated GC value.

Fig 3 caption. "The red horizontal line represents the significance threshold at 0.05. Bar plots (P value < 0.05)" These values are inconsistent with what's shown on the y axis, which is a more conventional threshold of around 10^{-6} .

Fig 5 (B). The AUC for PRS only should be indicated on the plots.

RESPONSE TO REVIEWERS

Reviewer #2 (Remarks to the Author)

1. The authors have conducted a multi-ethnicity, multi-trait meta-analysis for SLE, and presented results on TWAS, drug repurposing analysis and PRS. The study should add important information to the field of SLE genetics. Multi-trait analyses is based on an assumption that different autoimmune diseases may share the same association signals. In general, sharing a locus among autoimmune diseases is a common phenomenon but sharing the same association signal is not systematically examined. The authors conducted LD score regression to evaluate distances among the diseases and should partially address the question. I have some concerns if the major signal comes from other autoimmune diseases rather than SLE. For example, *CCDC88B* is a well-known locus that is associated with MS, alopecia areata, autoimmune thyroid diseases, PBC and Crohn's disease. In this case, I believe there should be a high threshold on result from SLE multi-ethnicity analysis before multi-trait analysis is conducted. If the SLE signal is low (may not be the case), and if the major signal comes from other autoimmune diseases, the validity of a SLE novel association would be in question. I hope the authors adopt a stringent standard in this part of analysis and clearly state it.

RESPONSE: Thank you for pointing this out. We did take a closer look into the 16 novel and replicable loci in multi-ancestry SLE-only GWAS. We found that three of these loci already attain genome-wide significant P values, while the other 13 loci all have P values $\leq 8.19 \times 10^{-5}$. In the case of *CCDC88B*, the P value for analyzing SLE alone is 1.46×10^{-6} , presenting suggestive evidence of association. A few other lines of evidence also support the association with SLE. By further assessing the strength and consistency of association signals of the lead SNP rs516124 across different participating studies after adjusting for ancestral differences of effect sizes, RATES further assigns a posterior probability of replicability of 0.997 to the signal.

As pointed out by the original manuscript that proposes MTAG, MTAG generates unbiased genetic effect estimates for each trait. We also observe that these estimates improve the risk prediction accuracy in our analysis, which provide further evidence of the validity of the results.

2. The couple of percentage improvement in PRS does not convince me the application value of PRS in this scenario. In what situations that PRS would help a physician make a decision? I believe the best chances are for individuals with the highest and lowest PRS, the two ends of the spectrum, and in situations that a decision is hard to make, for example based on ANA and anti-ds-DNA autoantibody alone. PRS scores will not help the individuals with PRS in the middle but if the authors can present a case that for those with extremely high or low PRS, this can clearly help with a clinical decision making, it will be great.

RESPONSE: Thank you for pointing this out! While the AUC improvement appears to be modest, the improvements are statistically significant (except for one scenario in MGI with P value of 0.065) when PRS is used in conjunction with clinical lab tests. However, the risk gradient curve showed a good stratification of

positive ANA and negative anti-dsDNA patients, the group that is difficult to diagnose. The prevalence of SLE in bottom, middle, and top quintiles are 9.91%, 15.09%, and 28.88%. The top quintile has 2.91 and 1.91 times more SLE cases than those in the bottom and middle and quintiles, respectively (Page 9, Paragraph 4). Individuals with extreme risk scores can be closely monitored for their progression and treated with possible intervention to slow down the progression.

3. TWAS results and drug repurposing: the analyses clearly provide useful information, but how to gauge the strength of the evidence? Could the connection go beyond merely a suggestive possibility for an available drug as efficacious for SLE?

RESPONSE: We can gauge the strength of the evidence from the CMap τ scores. We consider a gene set to have stronger connectivity with the drug if the drug more consistently reverses the expression levels of all genes in the gene set. τ score allows us to compare the strength of connectivity between different gene sets. For example, CMap τ score of -75 indicates that the drug more consistently reverses the expression level of the TWAS significant genes than 75% of all reference gene sets. While τ score can be used to assess the potential for repurposing a given drug for SLE treatment, additional validation is needed to confirm the finding.

Minor points:

4. line 189-196, the description of tissue specificity is little value comparing LCL and PBMC, I believe.

RESPONSE: Thank you for the suggestion! We have removed this paragraph from the manuscript.

5. Supplementary Table 6 may benefit the reader if it contains information on reported gene/closest gene(s).

RESPONSE: Thank you for pointing this out! We have added mapped target genes via Open Targets Genetics to the **Supplementary Table 6**.

Reviewer #3 (Remarks to the Author)

1. General comments: this is an important study, worthy of report in Nat Comm. I did not notice any major methodological flaws or other errors but the standard of write-up is not high enough. I think the manuscript would benefit from a thorough rewriting that eliminated excessive wording (I think about 25% could go), and improved the organisation. Particularly the sections in red (which appear to be edits from an earlier round of review in which I was not involved) are often verbose and not well integrated with the original text. To cite one very small issue, the authors switch between using “LD” and spelling out “linkage disequilibrium” breaking a basic rule, once you introduce an acronym (and LD was not defined) use it consistently. That’s a minor point but symptomatic of insufficient care. Most important is that too much of the wording is vague so that the exact meaning is fudged – I have specific examples below. Datasets are one important area for improvement, they should be described and named in one place and then referred to exactly by the specified name.

RESPONSE: Thank you! We have revised the manuscript to make it clearer, including the consistent use of acronyms after their definition and reorganizing the datasets description.

2. Some key parts of the analysis are not multi-ancestry, e.g. the selection of genetically-correlated traits. Because the title emphasises multi-ancestry, it should be more clear e.g. in a workflow diagram, what aspects of the analysis were not multi-ancestry.

RESPONSE: Thank you for pointing this out. We have updated the **Supplementary Figure 1** to indicate the analyses that only use samples of European ancestry (i.e., LDSC genetic correlation estimations) and the analyses that use multi-ancestry samples.

3. Regarding the genetically-correlated traits, significant at $FDR < 0.05$ is a weak criterion, I would have preferred an additional threshold on the estimated correlation coefficient to make the analysis more specific to SLE. Otherwise traits poorly-correlated with SLE can be collectively more important than SLE in the MTAG analysis.

RESPONSE: Thank you for pointing this out. We did take a closer look into the 16 novel and replicable loci in the multi-ancestry SLE-only GWAS. We found that three of these loci reaches genome-wide significant P values, while the other 13 loci have P values $\leq 8.19 \times 10^{-5}$, indicating possible association. It should also be noted that “MTAG generates trait-specific effect estimates for each SNP” (PMID: 29292387). In the article that proposes MTAG, the authors also noted that estimates from MTAG analysis improves the accuracy of genetic risk prediction of individual traits.

To further examine the impact of the choice of additional traits for analysis, we redid the MTAG analysis using traits whose genetic correlation with SLE have P values less than 0.05/91 (the Bonferroni threshold for testing

genetic correlations for each pair of 14 traits). There are 6 traits that satisfy the criteria, including rheumatoid arthritis, systemic sclerosis, primary biliary cirrhosis, type 1 diabetes, Sjogren's syndrome, and celiac disease. We plotted the P values of the sentinel variants across 106 loci (Figure below). We observed nearly complete agreement between the old and new multi-ancestry and multi-trait GWAS analyses with Pearson's correlation being .999. Our results demonstrate that using a stringent criterion to select phenotypes for MTAG analysis has minimal impact on the results.

4. Even based on this very large dataset, prediction accuracy using PRS is not high (as for many other complex conditions): the AUC is modest and shows only minimal improvement over current lab tests. A more honest discussion needs to be had of its limited utility: I can't foresee any clinical use for prediction at the accuracy shown here. If the authors disagree they should argue the case not just make vague claims lacking justification such as L371 "the use of PRS could facilitate early diagnosis and risk screening".

RESPONSE: Thank you for pointing this out! It is correct that AUC improvements are modest, yet they are statistically significant (except for one scenario in MGI with P value of 0.065) when PRS is used in conjunction with clinical lab tests.

On the other hand, the advantage of using PRS is better illustrated with the risk gradient curve, which plots the fraction of cases for each quintile of PRS distribution. The risk gradient curve shows that using PRS can further stratify patients with positive ANA and negative anti-dsDNA especially in BioVU database, the group that are most common and difficult to diagnose. The prevalence of SLE in bottom quintile of PRS distribution is 9.91%, that in the middle quintile is 15.09%, and that in the top quintile is 28.88%. The top quintile has 2.91 and 1.91 times more SLE cases than those in the bottom and middle quintiles, respectively (Page 9, Paragraph 4). Given the big differences in the risk between individuals at opposite extremes of the PRS distribution, it is important to consider closely monitoring their progression and treating them with possible intervention to slow down the progression.

To further clarify the clinical utility of PRS, in our manuscript, we mention “PRS would not replace conventional lab tests, but as germline DNA usually would not change over lifetime, the use of PRS could facilitate early diagnosis and risk screening. PRS has the most clinical utility in individuals with extreme PRS values, who will benefit from careful monitoring for progression. With the improvement in the understanding of polygenic architecture of complex diseases, the advancement of statistical methodology, and the increase of genetic diversity in genotyped samples, we could expect further improvements in the accuracy of genetic prediction.” (Page 10, Paragraph 3).

Specific points:

5. L43 “16 novel GWAS loci and an additional 22 novel loci” I think these are different definitions of “loci” and it would be better to indicate this with different words.

RESPONSE: Thank you for pointing this out. We have changed the wording to “16 novel GWAS loci and an additional 22 novel TWAS loci” (Page 2, Paragraph 1).

6. L58 “ranges from 20 to 150” is there any explanation for such a wide range? Are these for different sectors of the general population, e.g. different ancestry groups?

RESPONSE: Indeed, the wide range of the reported SLE prevalence can be due to varying geographic locations (countries, states), ancestry groups, study type, and study period. We have also adjusted the wordings of this sentence to reflect the SLE prevalence worldwide: “The prevalence of SLE worldwide was estimated to range from 37 to 123 cases per 100,000 individuals depending on geographic locations, ancestry groups, study type, and study period.” (Page 3, Paragraph 1).

7. L112 “significant SLE-correlated traits (false discovery rate < 0.05)” this text suggests phenotypic correlations but it becomes clear that the authors are talking about genetic correlations.

RESPONSE: We have adjusted the wording to “significantly genetically-correlated traits with SLE” (Page 4, Paragraph 2).

8. L113 “of significant genetic correlation with SLE” presumably this is referring to the 10 disorders in the previous sentence (now correctly described as genetically-correlated)? Much clearer to say that.

RESPONSE: Thank you for pointing this out! Indeed, we refer to traits genetically-correlated with SLE.

9. L123 “the linkage equilibrium-based pruning” better to call this “LD-based” as in L125

RESPONSE: We have changed it accordingly (Page 4, Paragraph 3).

10. L126 typo: r2

RESPONSE: We have fixed the wording to “the squared correlation between alleles (r^2) cutoff of 0.2” (Page 4, Paragraph 3).

11. L130 I know this term comes from other authors but “replicability” should not be used without details/explanation. This term is misleading and I would prefer it was avoided, but at least say “this term was introduced by X to mean ...”. Whether a hit is replicable depends on the details of the replication study, but most important is whether it is a valid association with effect size above some threshold. Instead “replicability” here is being used for a criterion of consistency of effect sizes implicitly assuming that consistency is an indicator of true effect – that needs an honest explanation, not just a misleading name.

RESPONSE: Thank you for the question! The term was defined by us in McGuire et al (PMID: 33785739). There, we define the “replicable” variants as the ones with genuine non-zero effects. So given a sufficiently powered replication study, e.g., one with large enough sample sizes from a matched population, we will be able to replicate the association signal under a significance threshold. By examining the strength and consistency of association signals across different participating studies in meta-analysis (after adjusting for genetic effect differences across ancestries), we can assign a “posterior probability of replicability” (PPR) to assess how likely the signal is genuine. Association signals that are genuine tend to have stronger and more consistent effects across studies and will tend to have higher PPR values. We have added the clarification of PPR to the text: “The term replicability was introduced by McGuire et al. that refers to variants with genuine non-zero effects. The posterior probability of replicability (PPR) quantifies how likely the signal is genuine and can be replicated in a sufficiently powered replication study, e.g., a large enough study from a matched population.” (Page 4 Paragraph 4).

12. L133 “posterior probability of replicability” this is even more confusing, it’s not even grammatical: whatever “replicability” means, it is not a binary term that can have a posterior probability. A statement that would be meaningful is “posterior probability that the true effect size exceeds X ...”. Again give some explanation of what this means, and it should be used in quotes to highlight that the term is misleading but attributed to other authors.

RESPONSE: Thank you for the question. As we discuss in the response to comment #11, we add the clarification of PPR to the text and cite the reference (Page 4 Paragraph 4).

13. L141 “these novel loci are all shared across ancestries” state the criterion for “shared”

RESPONSE: We apologize for the unclear description. We meant to refer to variants with homogeneous effects across ancestries. In the revised manuscript, we added that “all 16 novel loci do not yield significant P values using Cochran’s Q tests for heterogeneities under the Bonferroni threshold of 0.05/106.” (Page 5, Paragraph 1).

14. L148 “A portion of previously known loci” quantify this – how many are you aware of? “most of them are still marginally significant” quantify “most” and “marginally”. Currently this claim is too vague to be useful.

RESPONSE: We have added the following: “94 previously known loci did not reach genome-wide significance in our study, yet most of them (91 out of 94 loci) still have association P values < 0.05.” (Page 5, Paragraph 1).

15. L155 “95% of identified sentinel variants are non-coding. However, it should be noted that 79% of sentinel variant and its associated tag variants are located in the non-coding region.” I find this confusing – the new text needs to be better integrated with the original text.

RESPONSE: We have changed the wordings to: “79% of variants in the identified loci (as defined by sentinel variants and variants having $r^2 > 0.8$ to sentinel variants) are non-coding.” (Page 5, Paragraph 2).

16. L172 “7,028 and 9,260 significant gene expression prediction models” since a “model” is very general I don’t know what is being counted here: are these numbers of significant genes? Also, significant compared with what?

RESPONSE: We have updated the sentence to the following: “To assess the accuracy of prediction models, we calculate Spearman’s correlation coefficients between measured and predicted gene expression using nested cross validation as described in Khunsriraksakul et al and assess whether the Spearman’s correlation is significantly different from zero. Using PUMICE, we obtained 7,028 and 9,260 genes with Spearman’s correlation coefficients > 0.1 and the corresponding P values < 0.05 for GEUVADIS and DGN, respectively.” (Page 5, Paragraph 3 – Page 6, Paragraph 1).

17. L175 “optimally integrate eQTL data of European ancestry with a multi-ancestry GWAS dataset” The wording is too vague for me to understand: it’s not enough to bury details in the methods, you have to allow readers of the main text to understand the key ideas. What is being optimised? What is meant by “integrate”? Are the “eQTL data of European ancestry” the datasets described above and similarly is “a multi-ancestry GWAS dataset” the same GWAS dataset described above?

RESPONSE: Thank you for the suggestion! We have now clarified in the revised manuscript about the TESLA method we use to perform TWAS. A TWAS method integrates gene expression prediction models with GWAS datasets, by predicting gene expressions according to the gene expression prediction model in the GWAS dataset, and then testing for the association between predicted gene expression and phenotype.

TESLA is based on a meta-regression model, which integrates SLE GWAS summary statistics from multiple participating studies of different ancestries with gene expression prediction models based on samples of European ancestry in DGN or GEUVADIS. Using meta-regression, TESLA jointly models the genetic effects across ancestries so that we can borrow strength from shared effects between ancestries to optimize TWAS power. TESLA is more powerful than alternative strategies that leverage fixed effect meta-analysis results to perform TWAS and the methods that only integrates only ancestrally matched GWAS and eQTL datasets. We have added the clarification of TESLA method to the text: “Next, we conducted TWAS analysis using TESLA, which integrates our SLE GWAS summary statistics from multiple participating studies of different ancestries with gene expression prediction models based on samples of European ancestry in DGN or GEUVADIS. Specifically, TESLA uses meta-regression to jointly model the genetic effects across ancestries so that we can borrow strength from shared effects between ancestries to optimize TWAS power.” (Page 6, Paragraph 1).

18. L190 When reporting 40 and 38, remind us out of how many.

RESPONSE: We have removed this paragraph per suggestion from Reviewer #2.

19. L248 “which serves as a positive control” if you want to claim this you need to describe the experimental design and analysis that justify it. Deleting this phrase would make the sentence just acceptable in my view, but strictly the “Some” at the start of the sentence is so weak that nothing follows from it.

RESPONSE: We have removed the phrase and the sentence in the manuscript now reads “Some of the identified drug classes, including glucocorticoid receptor agonists, are already being commonly used in the clinic to treat SLE and supports the validity of these results.” (Page 7, Paragraph 3).

20. L260 “1) a large European GWAS” GWAS for what trait? Of what size? If for SLE, you need to state its overlap with the GWAS described here. For 2) and 3) I assume the datasets are those described above in this manuscript, but why introduce new terminology here? (Also I think “MA & MT” may confuse and better to write MAMT).

RESPONSE: We have revised our manuscript accordingly, i.e., “We analyzed three sets of GWAS summary statistics, including 1) SLE GWAS of European ancestry (N = 6,748 cases and 11,516 controls) (Ref), which we also include as part of MA and MAMT analyses, 2) MA for SLE, and 3) MAMT for SLE.” (Page 7, Paragraph 4 – Page 8, Paragraph 1). We have also updated the acronym “MA & MT” to “MAMT”.

21. L263 introduces two datasets not in this list. I find the \is presentation to be confusing.

RESPONSE: Thank you for pointing this out! We have added the following statement: “To validate the accuracy of PRS models, we use two independent biobanks that are linked to electronic medical records (EMRs), including

MGI (N = 34,702) and BioVU (N = 49,707), which are not part of the GWAS discovery cohorts.” (Page 8, Paragraph 1).

22. L269 What is a “model” here? Also, sample sizes for the datasets – and statement of their relationship to the primary dataset - should be given above, at first mention.

RESPONSE: We have clarified in the main text now that “model” refers to PRS model (Page 8, Paragraph 2). We have moved dataset description, sample size, and their relationship to the primary GWAS datasets to Page 8, Paragraph 1.

23. L278 “sparsity” how “sparse” is LASSOSUM? Also is it significantly better than other methods? Where are the reports of predictive accuracy and their SEs for all the PRS methods? You imply that LASSOSUM is the only sparse method, but pruning and thresholding is also sparse.

RESPONSE: We have now included a table showing the number of variants contained within each PRS model in **Supplementary Table 11**. LASSOSUM model contains 8,657 variants, while PRS-CS-auto model contains 877,453 variants.

In general, LASSOSUM did perform significantly better than other PRS methods (based on two-tailed Delong’s test). PRS-CS-auto performs comparably to LASSOSUM in most cases, but performing significantly worse in one scenario. We provide more detailed performance comparison in **Supplementary Table 10**.

We agree that pruning and thresholding is also a sparse method. We have fixed the description accordingly, i.e., “We found that LASSOSUM on average yielded more accurate models compared to other PRS methods” (Page 8, Paragraph 3).

24. L281 “performed the best” best compared with what? Are you just saying that MAMT beats MA? If so the superlative “best” is misleading.

RESPONSE: We have adjusted the wordings to the following: “Moreover, PRS based on GWAS summary statistics derived from MAMT almost always performed better than that of MA or Ref regardless of the PRS methods used.” (Page 8, Paragraph 3).

25. L285 “biobanks based on electronic medical records” As stated above, better to consolidate descriptions of datasets in one place not introduce in bits and pieces. Why is EMR data relevant to the decision to examine sensitivity?

RESPONSE: We have consolidated the following phrase to the paragraph describing the datasets. We also add the reasons why we performed sensitivity analyses in the text: “Since our external testing datasets are EMR-

based biobanks, which are susceptible to incorrect data entry or false positive diagnosis, we conducted sensitivity analyses to evaluate how the PRS models performed using SLE cases defined by three different algorithms.” (Page 8, Paragraph 4).

26. L316 “more likely to develop SLE” I accept that such probability statements are widely used, but unnecessary and I think we’d be better off if authors made a more direct statement of facts: you observed/estimated/computed that there were 3.8 times as many cases in ... than in ...

RESPONSE: We have changed our wording to: “We observed that there were 3.81 times as many cases in the top quintile than in the bottom quintile of the PRS distribution”. (Page 9, Paragraph 2).

27. L324 change “borderline significant” to “non-significant”. It seems disappointing that PRS gives so little advantage over the biomarkers alone, since the latter were said to have “low accuracy”. Also the biomarkers seem to add nothing to the AUC of PRS alone – this observation should be explicitly reported.

RESPONSE: We have changed the sentence from “Similarly, in MGI, the AUC of using PRS + ANA + anti-dsDNA is 0.75, which improves the AUC of using ANA + anti-dsDNA alone (0.74) with a borderline significant P value 0.065” to “Similarly, in MGI, the AUC of using PRS + ANA + anti-dsDNA is 0.75, which improves the AUC of using ANA + anti-dsDNA alone (0.74) with P value of 0.065”.

We have now added the AUCs of using PRS only to the figure. As we can see from the revised figure (**Figure 5B**), adding biomarkers to PRS significantly improve the AUC compared to using PRS alone (MGI: AUC of PRS = 0.65, AUC of PRS + ANA + anti-dsDNA = 0.75 with P value of 1.85×10^{-9} ; BioVU: AUC of PRS = 0.69, AUC of PRS + ANA + anti-dsDNA = 0.75 with P value of 8.84×10^{-9}).

28. L490 “produce a larger number of significant gene expression models” similar to comment above, “models” is ambiguous here and I can’t understand the claim.

RESPONSE: We have clarified “significant models” in the manuscript, i.e., “To assess the accuracy of prediction models, we calculate Spearman’s correlation coefficients between measured and predicted gene expression using nested cross validation as described in Khunsriraksakul et al. and assess whether the Spearman’s correlation is significantly different from zero⁴¹. Using PUMICE, we obtained 7,028 and 9,260 significant genes with Spearman’s correlation coefficients > 0.1 and the corresponding P values < 0.05 from GEUVADIS and DGN, respectively” (page 5 last paragraph). We have also changed the sentence in L490 of original manuscript to: “Thus, PUMICE can more accurately predict gene expression levels using genotype data as input compared to alternative approaches.” (Page 13, Paragraph 3).

29. Fig 1 (right) avoid using significance as a proxy for effect size as it can be misleading. Use significance as a threshold, but show estimated GC value.

RESPONSE: Thank you for pointing this out. The magnitude of genetic correlation is represented by the color on the line connecting two nodes. We have added the following statement to the figure legend: "The colors of the lines represent the magnitude of genetic correlation estimates using LDSC."

30. Fig 3 caption. "The red horizontal line represents the significance threshold at 0.05. Bar plots (P value < 0.05)" These values are inconsistent with what's shown on the y axis, which is a more conventional threshold of around 10^{-6} .

RESPONSE: Thank you for the suggestion! We have adjusted the color of the horizontal line on **Figure 3C** to blue so that it differs from that of **Figure 3A** and **3B**.

31. Fig 5 (B). The AUC for PRS only should be indicated on the plots.

RESPONSE: We have added the AUC of PRS only to **Figure 5B**.

REVIEWERS' COMMENTS

Reviewer #2 (Remarks to the Author):

My earlier point was that the strength of association signals for the novel loci needs to be strong for SLE alone. The authors demonstrated that the 16 loci they identified all have a P value smaller than $8.19E-5$. This addresses my concern on this issue.

The effect of combining PRS and autoantibody results is a good attempt. The improvement is not as convincing and should be duly acknowledged. The numbers between line 310 and 332 are all over the places, including 0.75-0.78; 0.74-0.79 with stringent SLE diagnosis criteria. The improvement, however, is from 0.73 to 0.75, and 0.74 to 0.75 (line 329-332). These are not improvement with any practical values (despite the statistical significance) and should be acknowledged.

Gene expression predicting models, on which TWAS was based, should be conducted, most ideally, using the most relevant cell types, as specific as possible, as gene-expression regulation can be cell-type specific as evidenced from numerous studies. In this sense, using TWAS findings to identify most relevant cell types for the disease seems backwards or falling into the trap of circular logic. I would like to hear the argument of the authors on this point.

Reviewer #3 (Remarks to the Author):

I think the authors for their responses to my previous comments. I don't fully agree with all their responses but I feel they have done enough and I don't wish to pursue any further points.

I just noticed the editing error "(Ref) on L 267.

RESPONSE TO REVIEWERS

Reviewer #2 (Remarks to the Author)

My earlier point was that the strength of association signals for the novel loci needs to be strong for SLE alone. The authors demonstrated that the 16 loci they identified all have a P value smaller than $8.19E-5$. This addresses my concern on this issue.

1. The effect of combining PRS and autoantibody results is a good attempt. The improvement is not as convincing and should be duly acknowledged. The numbers between line 310 and 332 are all over the places, including 0.75-0.78; 0.74-0.79 with stringent SLE diagnosis criteria. The improvement, however, is from 0.73 to 0.75, and 0.74 to 0.75 (line 329-332). These are not improvement with any practical values (despite the statistical significance) and should be acknowledged.

RESPONSE: Thank you for pointing this out! The improvement from 0.75 to 0.78 in MGI and 0.74 to 0.79 in BioVU (Lines 309-311) refer to the analysis where we compared the prediction performance when different case definitions were used (i.e., Def1, Def12). To clarify this, we have changed the sentence to the following: “When defining SLE cases using more stringent criteria (i.e., using Def 12 instead of Def 1), we improved the AUC of PRS from 0.75 to 0.78 in MGI and 0.74 to 0.79 in BioVU”

For lines 329-332, this analysis was performed in a subset of patients who have available information on both ANA and anti-dsDNA tests to compare the prediction performance between “ANA + anti-dsDNA” and “PRS + ANA + anti-dsDNA.” We agree with the reviewer that the AUC improvement is modest albeit statistically significant (except for one scenario in MGI with P value of 0.065) when PRS is used in conjunction with clinical lab tests.

In the article, we emphasize that the benefits of PRS lie in its utility to stratify individuals for their risk of progressing to SLE. Specifically, the risk gradient curve showed a good stratification of individuals with positive ANA and negative anti-dsDNA levels, the group that is most difficult to diagnose. The prevalence of SLE in the bottom, middle, and top quintiles are 9.91%, 15.09%, and 28.88%. Individuals in the top quintile have 2.91 and 1.91 times elevated SLE risks compared to those in the bottom and middle and quintiles, respectively (Page 9, Paragraph 4). Individuals with extreme risk scores can be closely monitored for their progression and treated with possible intervention to slow down the progression.

2. Gene expression predicting models, on which TWAS was based, should be conducted, most ideally, using the most relevant cell types, as specific as possible, as gene-expression regulation can be cell-type specific as evidenced from numerous studies. In this sense, using TWAS findings to identify most relevant cell types for the disease seems backwards or falling into the trap of circular logic. I would like to hear the argument of the authors on this point.

RESPONSE: We totally agree with the reviewer that TWAS ideally should be performed using the most relevant tissues or cell types. In our analysis, we selected whole blood and lymphoblastoid cell line to conduct TWAS analysis based on our existing knowledge, because SLE is an autoimmune disease that involves dysfunctional B cells, T cells, and potentially innate immune system (PMID: 31037070). Since whole blood “tissue” is composed of many distinct cell types, we were interested in the relative contribution and importance of each cell type to the pathogenesis of SLE. Therefore, we performed cell type enrichment analysis using DICE dataset and B cell dataset (from SLE subjects) as a reference to find relevant cell types.

Reviewer #3 (Remarks to the Author)

I think the authors for their responses to my previous comments. I don't fully agree with all their responses but I feel they have done enough and I don't wish to pursue any further points.

1. I just noticed the editing error "(Ref) on L 267.

RESPONSE: This "Ref" refers to the set of GWAS summary statistic we used to compare with MA and MAMT analyses.